

# A single-column particle-resolved model for simulating the vertical distribution of aerosol mixing state: WRF-PartMC-MOSAIC-SCM v1.0

Jeffrey H. Curtis[1], Nicole Riemer[1], and Matthew West[2]

[1]Department of Atmospheric Sciences, University of Illinois at Urbana-Champaign, 105 S Gregory St., Urbana, IL 61801, USA
[2]Department of Mechanical Science and Engineering, University of Illinois at Urbana-Champaign, 1206 W. Green St., Urbana, IL 61801, USA

*Correspondence to:* N. Riemer
(nriemer@illinois.edu)

**Abstract.** The particle-resolved aerosol model PartMC-MOSAIC was previously developed to predict the aerosol mixing state as it evolves in the atmosphere. However, the modeling framework was limited to a 0-D box model approach without resolving spatial gradients in aerosol concentrations. This paper presents the development of stochastic particle methods to simulate turbulent diffusion and dry deposition of aerosol particles in a vertical column within the planetary boundary layer. The new

model, WRF-PartMC-MOSAIC-SCM, resolves the vertical distribution of aerosol mixing state. We verified the new algorithms with analytical solutions for idealized testcases and illustrate the capabilities with results from a two-day urban scenario that shows the evolution of black carbon mixing state in a vertical column.

## 1 Introduction

Aerosol particles impact the Earth's radiative budget directly by scattering and absorbing shortwave radiation (McCormick and
10 Ludwig, 1967; Charlson and Pilat, 1969; Charlson et al., 1992), and indirectly by modifying cloud microphysical properties (Twomey, 1977; Albrecht, 1989; Rosenfeld, 2000). The magnitude of these impacts on climate depends not only on the bulk amount of aerosol material in the atmospheric column, but also on its vertical distribution within the column (Haywood and Shine, 1997; Schulz et al., 2006; Zarzycki and Bond, 2010; Samset and Myhre, 2011; Ban-Weiss et al., 2012), and on the microphysical characteristics of the aerosol population, such as the size distribution of the particles and the aerosol composition
on a per-particle level (McFiggans et al., 2006; Moffet and Prather, 2009; Zelenyuk and Imre, 2009; Zelenyuk et al., 2010). For the purposes of this paper we use the term "aerosol mixing state" to refer to the distribution of chemical species across the aerosol population (Riemer and West, 2013).

Observational evidence shows that aerosol mixing state varies with altitude. For example, Pratt and Prather (2010) used the aircraft aerosol time-of-flight mass spectrometer to measure vertical profiles of single-particle composition over Wyoming and
20 northern Colorado and found that carbonaceous particles were mixed with ammonium, nitrate and sulfate at low altitudes, while





they were mixed with sulfate and sulfuric acid at higher altitudes. Measurements with the single-particle soot photometer (SP2) showed that the fraction of black carbon particles that are heavily coated increases with altitude (McMeeking et al., 2011).

These experimental findings confirm that the composition of individual aerosol particles constantly changes during the particles' lifetime as a result of aging processes such as coagulation (Fassi-Fihri et al., 1997), condensation (Pósfai et al., 1999)

and photochemical processes (Kotzick and Nießner, 1999), and that this is intimately linked to the transport processes in the atmosphere. To better represent these processes in chemical transport models, several two-dimensional sectional models have been developed, such as MADRID-BC (Oshima et al., 2009b, a), the MS-Resolved WRF-Chem (Matsui et al., 2013), and WRF-Chem/ATRAS-MOSAIC (Matsui et al., 2014). A common feature of these models is that they use a two-dimensional sectional framework to represent black-carbon-containing particles, with one dimension being dry diameter and the other dimension

being black carbon mass fraction. Building on previous two-dimensional sectional frameworks, the MOSAIC-MIX model (Ching et al., 2016) adds an additional dimension to represent hygroscopicity and shows that this optimizes the calculations of CCN concentrations and aerosol optical properties. The SCRAM model (Zhu et al., 2015, 2016), also a two-dimensional sectional model, uses an alternative discretization based on both size and composition where composition is tracked by mass fractions of different chemical groups such as inorganic hydrophilic, organic hydrophilic, organic hydrophobic, black carbon,

and dust.

From the application of the different types of aerosol models described above within spatially-resolved 3D chemical transport models we learn that it is important to track the aerosol mixing state in order to accurately predict particle aging, the associated aerosol optical properties, and the resulting heating rates (Riemer et al., 2003; Matsui et al., 2013; Zhang et al., 2014; Matsui, 2016; Zhu et al., 2016). Such 3D chemical transport models that resolve additional mixing state information are focused on

black carbon. Further extension of aerosol bin schemes to include additional dimensions to capture greater mixing state detail eventually becomes computationally prohibitive.

In contrast to the distribution-based models mentioned here, particle-resolved aerosol models simulate a representative group of particles distributed in composition space, treating coagulation, condensation/evaporation, and other important processes on an individual particle level. Relative particle positions within this computational volume are not tracked but instead processes

such as coagulations are simulated stochastically, following the approach pioneered by Gillespie (1975). Particle methods are attractive, because they resolve the full aerosol mixing state without any ad hoc assumptions. The storage cost of these models is proportional to the number of particles, the computational cost for evaporation/condensation is proportional to the number of particles, and the computational cost for coagulation is proportional to the number of coagulation events (Riemer et al., 2009).

For the large number of computational particles needed for atmospheric simulations, we developed efficient algorithms for

coagulation (Riemer et al., 2009; Michelotti et al., 2013) and for appropriately weighting computational particles (DeVille et al., 2011). These were implemented it in the Particle Monte Carlo (PartMC) model for simulating atmospheric aerosol dynamics and coupled with the state-of-the-art aerosol chemistry model MOSAIC (Zaveri et al., 2008), which simulates the gas- and particle-phase chemistries, particle-phase thermodynamics, and dynamic gas-particle mass transfer in a deterministic manner. The coupled model system, PartMC-MOSAIC, predicts number, mass, and full composition distributions, and is therefore

suited for applications where any or all of these quantities are required. The particle-resolved approach eliminates any errors





associated with artificial numerical diffusion in composition space. As a result, its treatment of aerosol mixing state dynamics and chemistry makes PartMC-MOSAIC suitable for use as a numerical benchmark of mixing state for more approximate models (McGraw et al., 2008; Kaiser et al., 2014).

In previous work PartMC-MOSAIC has been used as a box model (Zaveri et al., 2010; Tian et al., 2014; Fierce et al., 2016; Ching et al., 2016), and hence it was not possible to resolve spatial gradients in aerosol mixing state. To overcome this limitation, we have now coupled PartMC-MOSAIC with the Weather Research and Forecast (WRF) model to allow transport of aerosol particle populations and gas species concentrations. In this paper we present the model development that couples PartMC-MOSAIC with the WRF Single Column Model, resulting in a fully-coupled 1D atmospheric-dynamics/aerosol-particle model that not only resolves the particle mixing state on a per-particle level but also resolves the vertical structure of the atmosphere.

This paper is structured as follows. In Sec. 2 we write the governing equations for the coupled gas-aerosol 1D column model. In Sec. 3 we discuss the specifics of the coupled model including numerical approximations to model the vertical transport of aerosols. In Sec. 4 we present test-case verification of the two new model processes of turbulent transport and particle removal by dry deposition. Section 5 shows the multidimensional particle-resolved results for an idealized scenario, focusing on the evolution of black-carbon-containing particles.

## 2   Coupled aerosol-gas governing equations

In this section we describe the model equations that govern the evolution of aerosol particles and trace gases in a vertical column. We include gas phase chemistry, gas-to-particle conversion, coagulation of aerosol particles, emission of aerosol and gases, and the transport of aerosol particles and trace gases in the vertical column. We ignore horizontal diffusion and advection of trace gases into and out of the column by assuming horizontal homogeneity.

An aerosol particle contains mass $\mu_a \geq 0$ of species $a$, for $a = 1, \ldots, A$, so that the particle composition is described by the $A$-dimensional vector $\boldsymbol{\mu} \in \mathbb{R}^A$. The cumulative aerosol number distribution at height $z$ with constituent masses $\boldsymbol{\mu}$ at time $t$ is $N(z, \boldsymbol{\mu}, t)$ (m$^{-3}$). The aerosol number distribution at height $z$ and time $t$ with constituent masses $\boldsymbol{\mu}$ is then defined by

$$n(z, \boldsymbol{\mu}, t) = \frac{\partial^A N(z, \boldsymbol{\mu}, t)}{\partial \mu_1 \partial \mu_2 \ldots \partial \mu_A} \tag{1}$$

with units m$^{-3}$ kg$^{-A}$.

The concentration of gas phase species $i$ at height $z$ and time $t$ is given by $g_i(z, t)$, for $i = 1, \ldots, G$, so that gas phase concentrations form the $G$-dimensional vector $\boldsymbol{g}(z, t) \in \mathbb{R}^G$. We assume that the first $C$ aerosol and gas species undergo gas-to-particle conversion and are indexed in the same order so that gas species $i$ partitions with aerosol species $i$ for $i = 1, \ldots, C$. Additionally, species $C + 1$ is assumed to be water.





The evolution of the multidimensional aerosol number distribution is given by

$$
\underbrace{\frac{\partial n(z,\boldsymbol{\mu},t)}{\partial t} + w(t,z)\frac{\partial n(z,\boldsymbol{\mu},t)}{\partial z}}_{\text{vertical advection}} - \underbrace{\left(\frac{\partial}{\partial z}\left(K_{\text{h}}(z,t)\rho_{\text{dry}}(z,t)\frac{\partial}{\partial z}\left(\frac{n(z,\boldsymbol{\mu},t)}{\rho_{\text{dry}}(z,t)}\right)\right)\right)}_{\text{turbulent transport}}
$$

$$
= \underbrace{\frac{1}{2}\int_0^{\mu_1}\int_0^{\mu_2}\cdots\int_0^{\mu_A}K(\boldsymbol{\mu}',\boldsymbol{\mu}-\boldsymbol{\mu}')n(z,\boldsymbol{\mu}',t)n(z,\boldsymbol{\mu}-\boldsymbol{\mu}',t)d\mu_1'd\mu_2'\dots d\mu_A'}_{\text{coagulation gain}}
$$

$$
- \underbrace{\int_0^{\infty}\int_0^{\infty}\cdots\int_0^{\infty}K(\boldsymbol{\mu},\boldsymbol{\mu}')n(z,\boldsymbol{\mu},t)n(z,\boldsymbol{\mu}',t)d\mu_1'd\mu_2'\dots d\mu_A'}_{\text{coagulation loss}} + \underbrace{\dot{n}_{\text{emit}}(z,\boldsymbol{\mu},t)}_{\text{emission}} \tag{2}
$$

$$
- \underbrace{\sum_{i=1}^{C}\frac{\partial}{\partial\mu_i}(c_a I_a(\boldsymbol{\mu},\boldsymbol{g},t)n(z,\boldsymbol{\mu},t))}_{\text{gas-particle transfer}} - \underbrace{\frac{\partial}{\partial\mu_{C+1}}(c_{\text{w}}I_{\text{w}}(\boldsymbol{\mu},\boldsymbol{g},t)n(z,\boldsymbol{\mu},t)}_{\text{water transfer}}
$$

$$
+ \underbrace{\frac{1}{\rho_{\text{dry}}(z,t)}\frac{\partial\rho_{\text{dry}}(z,t)}{\partial t}n(z,\boldsymbol{\mu},t)}_{\text{air density change}},
$$

where $w(z,t)$ (m s$^{-1}$) is the vertical velocity, $K_{\text{h}}(z,t)$ (m$^2$ s$^{-1}$) is the diffusion coefficient of heat, $K(\boldsymbol{\mu},\boldsymbol{\mu}')$ (m$^3$ s$^{-1}$) is the coagulation rate between particles $\boldsymbol{\mu}$ and $\boldsymbol{\mu}'$, $\dot{n}_{\text{emit}}(z,\boldsymbol{\mu},t)$ (m$^{-3}$ kg$^{-A}$ s$^{-1}$) is the number distribution rate of aerosol emissions,

5    $c_a$ (kg mol$^{-1}$) is the conversion factor from moles of gas species $a$ to aerosol species $a$, $I_a(\boldsymbol{\mu},\boldsymbol{g},t)$ (mol s$^{-1}$) is the condensation flux of gas species $a$, and $c_{\text{w}}$ (kg mol$^{-1}$) is the conversion factor for water, and $I_w(\boldsymbol{\mu},\boldsymbol{g},t)$ (mol s$^{-1}$) is the condensation flux for water. The turbulent transport term is written using the gradient of mixing ratio rather than the gradient of concentration to account for the vertical variations in density that are present in the atmosphere (Equation (6), Venkatram (1993)). Equation (2) does not contain a term for gravitational sedimentation since we focus our test case on submicron particles for which the

10    settling velocities are very small. As an example, over the course of the 48-hour simulation period a 1 μm particle would only settle by about 10 m, which is less than the smallest vertical grid size used here. Gravitational settling should be included in scenarios that involve larger particles such as sea salt and dust or simulations with finer vertical resolution.



The evolution of trace gas concentrations is given by

$$
\frac{\partial g_i(z,t)}{\partial t} + \underbrace{w(z,t)\frac{\partial g_i(z,t)}{\partial z}}_{\text{vertical advection}} - \underbrace{\left( \frac{\partial}{\partial z}\left( K_{\mathrm{h}}(z,t)\rho_{\mathrm{dry}}(z,t)\frac{\partial}{\partial z}\left( \frac{g_i(z,t)}{\rho_{\mathrm{dry}}(z,t)} \right) \right) \right)}_{\text{turbulent transport}}
$$

$$
= \underbrace{\dot{g}_{\mathrm{emit},i}(z,t)}_{\text{emission}} + \underbrace{R_i(\boldsymbol{g}(z,t))}_{\text{chemical reactions}} + \underbrace{\frac{1}{\rho_{\mathrm{dry}}(z,t)}\frac{\partial\rho_{\mathrm{dry}}(z,t)}{\partial t}g_i(z,t)}_{\text{air density change}}
$$

$$
- \underbrace{\int_0^\infty \int_0^\infty \cdots \int_0^\infty I_i(\boldsymbol{\mu},\boldsymbol{g},t)n(z,\boldsymbol{\mu},t)d\mu_1 d\mu_2 \dots d\mu_A}_{\text{gas-particle transfer}},
$$

(3)

where $\dot{g}_{\mathrm{emit},i}(z,t)$ (mol m$^{-3}$ s$^{-1}$) is the emission rate of gas species $i$ and $R_i(\boldsymbol{g}(z,t))$ (mol m$^{-3}$ s$^{-1}$) is the concentration growth rate of gas species $i$ due to gas-phase chemical reactions.

For Eq. (2) and Eq. (3) we use reflective boundary conditions at the top of the domain and partly reflecting, partly absorbing boundary conditions at the surface. For the aerosol distribution in Eq. (2), this is given at the top of the domain ($z = h$) by

$$
K_{\mathrm{h}}(z,t)\rho_{\mathrm{dry}}(z,t)\frac{\partial}{\partial z}\left( \frac{n(z,\boldsymbol{\mu},t)}{\rho_{\mathrm{dry}}(z,t)} \right) = 0 \text{ at } z = h,
$$

(4)

and at the surface by

$$
K_{\mathrm{h}}(z,t)\rho_{\mathrm{dry}}(z,t)\frac{\partial}{\partial z}\left( \frac{n(z,\boldsymbol{\mu},t)}{\rho_{\mathrm{dry}}(z,t)} \right) = V_{\mathrm{d}}(\boldsymbol{\mu})n(z,\boldsymbol{\mu},t) \text{ at } z = 0.
$$

(5)

Here $V_{\mathrm{d}}(\boldsymbol{\mu})$ (m s$^{-1}$) is the dry deposition velocity, which depends on particle size and composition. Dry deposition velocities for aerosols are computed using the size-dependent dry deposition scheme described in Zhang et al. (2001) by their equations (1), (2), (3), (5), (6), (7c), (8), and (9). Instead of using Equation (4) in Zhang et al. (2001), which describes the calculation of the aerodynamic resistance, the aerodynamic resistance computed by the WRF model is used, as described by McRae et al. (1982). Further, we do not need to use the parameterization for the correction of particle size for high relative humidity

conditions given in equation (10) in Zhang et al. (2001), since we explicitly compute the water content of the aerosol particles and hence directly calculate the particles' wet diameters.

For the gas phase concentrations of Eq. (3) the boundary condition at the top of the domain is given by

$$
K_{\mathrm{h}}(z,t)\rho_{\mathrm{dry}}(z,t)\frac{\partial}{\partial z}\left( \frac{g_i(z,t)}{\rho_{\mathrm{dry}}(z,t)} \right) = 0 \text{ at } z = h,
$$

(6)

and at the surface by

$$
K_{\mathrm{h}}(z,t)\rho_{\mathrm{dry}}(z,t)\frac{\partial}{\partial z}\left( \frac{g_i(z,t)}{\rho_{\mathrm{dry}}(z,t)} \right) = V_{\mathrm{d},i}\, g_i(z,t) \text{ at } z = 0,
$$

(7)

where $V_{\mathrm{d},i}$ (m s$^{-1}$) is the dry deposition velocity of gas species $i$. The dry deposition velocity of each gas species is determined by WRF/Chem as described in Grell et al. (2005) with the use of the surface resistance parameterization from Wesely (1989).





## 3  Model discretization

We coupled the different model components (WRF, PartMC, and MOSAIC) by using the operator splitting (Press et al., 2007, section 20.3.3)

$$\Phi_{\Delta t} = \Phi_{\Delta t}^{\text{WRF}} \circ \Phi_{\Delta t}^{\text{PartMC}} \circ \Phi_{\Delta t}^{\text{MOSAIC}} \circ \Phi_{\Delta t}^{\text{Trans}}, \tag{8}$$

which allows for the use of independent numerical methods to solve each portion.

The Advanced Research Weather Research and Forecasting (WRF-ARW) Model ($\Phi_{\Delta t}^{\text{WRF}}$) is used to solve for the meteorological variables, details of which are further described in Skamarock et al. (2008). WRF computes temperature, pressure, eddy diffusivity, aerodynamic resistance and dry deposition velocity for gases, which are then used in Eqs. (2) and (3). The Particle-resolved Monte Carlo (PartMC) model ($\Phi_{\Delta t}^{\text{PartMC}}$) is used to treat the coagulation term and the emission term in Eq. (2).

This is done with a stochastic approach as described in Riemer et al. (2009), DeVille et al. (2011), and Michelotti et al. (2013).

The Model for Simulating Aerosol Interactions and Chemistry (MOSAIC) ($\Phi_{\Delta t}^{\text{MOSAIC}}$) is used to solve gas-phase and gas-to-particle chemistry terms in Eqs. (2) and (3). The composition of each particle can change due to evaporation and condensation of chemical species to and from the gas phase. The MOSAIC model (Zaveri et al., 2008) determininistically treats gas-phase chemistry and gas-particle partitioning. MOSAIC consists of the gas-phase mechanism Carbon-Bond Mechanism version Z

(CBM-Z) (Zaveri and Peters, 1999), the Multicomponent Equilibrium Solver for Aerosols (MESA) for aerosol solid-liquid partitioning (Zaveri et al., 2005a), the Multicomponent Taylor Expansion Method (MTEM) for estimating activity coefficients of electrolytes and ions in aqueous solutions (Zaveri et al., 2005b) and the Adaptive Step Time-split Euler Method (ASTEM) for gas-particle partitioning (Zaveri et al., 2008). MOSAIC uses the Secondary Organic Aerosol Model (SORGAM) scheme (Schell et al., 2001) for the treatment of secondary organic aerosol. The transport model ($\Phi_{\Delta t}^{\text{Trans}}$) solves the turbulent transport

terms for gas species and particles, deterministically for gases and stochastically for particles as described in Sec. 3.1. The finite volume method used for particle transport is also applied to gas transport to provide consistency between the transport of gases and aerosols.

### 3.1  Stochastic aerosol transport algorithm for turbulent diffusion

This section details the treatment of the turbulent transport term of Eq. (2). Within each grid cell $k$ of the model domain we

represent the aerosol state $\Pi_k$ with $N_{\text{p}}^k$ particles, where $\Pi_k = \{\boldsymbol{\mu}^1, \boldsymbol{\mu}^2, \dots, \boldsymbol{\mu}^{N_{\text{p}}^k}\}$, and the particle order is not significant. There may be multiple identical particles in a single grid cell, so $\Pi_k$ is a multiset in the sense of Knuth (1998, p. 473). Each particle is an $A$-dimensional vector $\boldsymbol{\mu}^i \in \mathbb{R}^A$ so that $\mu_a^i$ is the mass of species $a$ in particle $i$, for $a = 1, \dots, A$ and $i = 1, \dots, N_{\text{p}}^k$.

Note that we do not simulate and track every particle within a grid cell. Rather, the aerosol population $\Pi_k$ can be thought of as populating a computational volume $V_k$ which is smaller than the physical volume of the grid cell. The computational volume

is representative of the mean properties of that grid cell. This assumption is the same as the one used for the box model version of PartMC-MOSAIC presented in Riemer et al. (2009), which simulated particles within a given computational volume that was representative of the well-mixed boundary layer during the day, and of the residual layer during the night. For simplicity we use here a flat weighting for all computational particles.



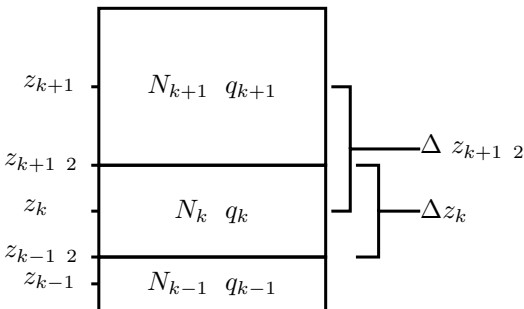

**Figure 1.** Schematic of the single column domain centered on grid cell $k$ with neighboring grid cells $k-1$ and $k+1$.

Since PartMC-MOSAIC resolves a finite population of particles $\Pi_k$ in a given volume $V_k$ within grid cell $k$, we must determine the finite number of particles that are transported in and out of grid cell $k$ due to turbulent transport. To determine the set of particles to remove from particle set $\Pi_k$ as well as to add to $\Pi_k$ from neighboring grid cells, we discretize the vertical turbulent transport term of Eq. (2) with respect to space, time, and particle number.

We first present the discretization in space and time in terms of deterministic number concentrations and particle number (Sec. 3.1.1). The resulting equation for turbulent transport of determinstic particle number is then discretized for stochastic particle number as shown in Sec. 3.1.2, the algorithm for particle sampling is given in Sec. 3.1.3, and time step selection is presented in Sec. 3.1.4

### 3.1.1   Discretization in space and time in terms of deterministic particle number

Figure 1 introduces our notation for the spatial discretization of the single column model domain. The vertical grid spacing $\Delta z$ is non-uniform and also varies over time. The variation over time is a result of WRF using the pressure-based vertical coordinate $\eta$, while the physical height $z$ of grid cells is computed from geopotential height.

To account for the variation in $\Delta z$ both with respect to height and to time, we define the distance from the top to the bottom edge of grid cell $k$ at time $t^s$ to be

$$\Delta z_k^s = z_{k+\frac{1}{2}}^s - z_{k-\frac{1}{2}}^s, \tag{9}$$

and the distance between the center of grid cells $k$ and $k+1$ at $t^s$ by

$$\Delta z_{k+\frac{1}{2}}^s = z_{k+1}^s - z_k^s. \tag{10}$$

To obtain transported number concentrations and eventually a discrete number of transported particles, we define the total aerosol number concentration $N(z,t)$ at height $z$ and time $t$ as

$$N(z,t) = \int_0^\infty \int_0^\infty \cdots \int_0^\infty n(z,\boldsymbol{\mu},t)\,d\mu_1 d\mu_2 \cdots d\mu_A, \tag{11}$$





where $n(z, \boldsymbol{\mu}, t)$ is the multidimensional aerosol number distribution. We then apply the turbulent transport process in terms of total number concentration as

$$\frac{\partial N(z,t)}{\partial t} = \frac{\partial}{\partial z}\left(K_{\mathrm{h}}(z,t)\rho(z,t)\frac{\partial q(z,t)}{\partial z}\right), \tag{12}$$

where $K_{\mathrm{h}}$ is eddy diffusivity of heat, $\rho(z,t)$ is density of dry air, and $q(z,t)$ is the mixing ratio defined by

$$5 \quad q(z,t) = \frac{N(z,t)}{\rho(z,t)}. \tag{13}$$

We consider a finite-volume discretization of Eq. (12) where we seek a solution for the cell average total number concentration $N_k(t)$ defined as

$$N_k(t) = \frac{1}{\Delta z_k}\int\limits_{z_{k-\frac{1}{2}}}^{z_{k+\frac{1}{2}}} N(z,t)dz. \tag{14}$$

Similarly we can define cell average mixing ratios by

$$10 \quad q_k(t) = \frac{1}{\Delta z_k}\int\limits_{z_{k-\frac{1}{2}}}^{z_{k+\frac{1}{2}}} q(z,t)dz. \tag{15}$$

Applying a finite volume approach to Eq. (12) and assuming a fixed grid within a time step yields

$$\int\limits_{z_{k-\frac{1}{2}}}^{z_{k+\frac{1}{2}}} \frac{\partial N(z,t)}{\partial t}dz = \int\limits_{z_{k-\frac{1}{2}}}^{z_{k+\frac{1}{2}}} \frac{\partial}{\partial z}\left(K_{\mathrm{h}}(z,t)\rho(z,t)\frac{\partial q(z,t)}{\partial z}\right)dz. \tag{16}$$

This can be transformed to give the flux form

$$\frac{\partial}{\partial t}N_k(t)\Delta z_k = \left.\left(K_{\mathrm{h}}(z,t)\rho(z,t)\frac{\partial q(z,t)}{\partial z}\right)\right|_{z_{k+\frac{1}{2}}} - \left.\left(K_{\mathrm{h}(z,t)}\rho(z,t)\frac{\partial q(z,t)}{\partial z}\right)\right|_{z_{k-\frac{1}{2}}}. \tag{17}$$

15   We numerically approximate the spatial derivative in Eq. (17) by

$$\left.\frac{\partial q(z,t)}{\partial z}\right|_{z=z_{k+\frac{1}{2}}} \approx \frac{q_{k+1}(t) - q_k(t)}{\Delta z_k}, \tag{18}$$

and the time derivative by

$$\left.\frac{\partial N_k(t)}{\partial t}\right|_{t=t^s} \approx \frac{N_k^{s+1} - N_k^s}{\Delta t}, \tag{19}$$

where $\Delta t = t^{s+1} - t^s$ is the constant time step. Let $N_k^s$ be the approximate solution of $N_k(t^s)$ and $q_k^s$ be the approximate

20   solution of $q_k(t^s)$, which satisfy the following fully discrete equation of

$$\frac{\left(N_k^{s+1} - N_k^s\right)\Delta z_k^s}{\Delta t} = \left(K_{\mathrm{h},k+\frac{1}{2}}^s \rho_{k+\frac{1}{2}}^s \frac{q_{k+1}^s - q_k^s}{\Delta z_{k+\frac{1}{2}}^s}\right) - \left(K_{\mathrm{h},k-\frac{1}{2}}^s \rho_{k-\frac{1}{2}}^s \frac{q_k^s - q_{k-1}^s}{\Delta z_{k-\frac{1}{2}}^s}\right). \tag{20}$$





We then replace the mixing ratio terms in Eq. (20) using the relationship $q_k^s = \frac{N_k^s}{\rho_k^s}$, which results in

$$\frac{\left(N_k^{s+1} - N_k^s\right)\Delta z_k^s}{\Delta t} = \left(K_{\mathrm{h},k+\frac{1}{2}}^s \rho_{k+\frac{1}{2}}^s \frac{\frac{N_{k+1}^s}{\rho_{k+1}^s} - \frac{N_k^s}{\rho_k}}{\Delta z_{k+\frac{1}{2}}^s}\right) - \left(K_{\mathrm{h},k-\frac{1}{2}}^s \rho_{k-\frac{1}{2}}^s \frac{\frac{N_k^s}{\rho_k^s} - \frac{N_{k-1}^s}{\rho_{k-1}}}{\Delta z_{k-\frac{1}{2}}^s}\right). \tag{21}$$

Given that we seek to represent the loss of particles from each grid cell to the neighboring cells and the gain of particles from those neighboring cells, we need a form of the equation in terms of gains and losses rather than a flux formulation. We rearrange Eq. (21) to isolate gain and loss terms of the number concentration $N_k^s$, giving

$$N_k^{s+1} = N_k^s + \underbrace{\underbrace{\left(\frac{\Delta t}{\Delta z_k^s \Delta z_{k+\frac{1}{2}}^s} \frac{\rho_{k+\frac{1}{2}}^s}{\rho_{k+1}^s} K_{\mathrm{h},k+\frac{1}{2}}^s\right)}_{\beta_{k+1,k}^{\mathrm{G},s}} N_{k+1}^s}_{N_{k+1,k}^{\mathrm{G},s}} - \underbrace{\underbrace{\left(\frac{\Delta t}{\Delta z_k^s \Delta z_{k+\frac{1}{2}}^s} \frac{\rho_{k+\frac{1}{2}}^s}{\rho_k^s} K_{\mathrm{h},k+\frac{1}{2}}^s\right)}_{\beta_{k,k+1}^{\mathrm{L},s}} N_k^s}_{N_{k,k+1}^{\mathrm{L},s}}$$

$$- \underbrace{\underbrace{\left(\frac{\Delta t}{\Delta z_k^s \Delta z_{k-\frac{1}{2}}^s} \frac{\rho_{k-\frac{1}{2}}^s}{\rho_k^s} K_{\mathrm{h},k-\frac{1}{2}}^s\right)}_{\beta_{k,k-1}^{\mathrm{L},s}} N_k^s}_{N_{k,k-1}^{\mathrm{L},s}} + \underbrace{\underbrace{\left(\frac{\Delta t}{\Delta z_k^s \Delta z_{k-\frac{1}{2}}^s} \frac{\rho_{k-\frac{1}{2}}^s}{\rho_{k-1}^s} K_{\mathrm{h},k-\frac{1}{2}}^s\right)}_{\beta_{k-1,k}^{\mathrm{G},s}} N_{k-1}^s}_{N_{k-1,k}^{\mathrm{G},s}}. \tag{22}$$

The four transport terms are illustrated in Fig. 2. The arrows in Fig. 2 show the transported number concentrations from and to grid cell $k$. The first subscript indicates the origin grid cell, and the second subscript indicates the destination grid cell. The superscripts indicate either gain (G) for the destination grid cell or loss (L) for the origin grid cell at time step $s$. For example $N_{k+1,k}^{\mathrm{G},s}$ is the number concentration transported from grid cell $k+1$ to grid cell $k$ at time $s$, representing a gain for grid cell $k$. $N_{k,k+1}^{\mathrm{L},s}$ is the number concentration transported from grid cell $k$ to $k+1$ at time $s$, representing a loss for grid cell $k$. These transport terms can also be expressed in terms of the product of a coefficient $\beta$ and the number concentration in the origin grid cell.

Eventually, we want to perform turbulent transport of discrete particles, therefore as the next step we express Eq. (22) in terms of real-valued particle number instead of number concentration. The deterministic real-valued particle number of grid cell $k$ at time $s$ is related to number concentration by

$$C_k^s = V_k^s N_k^s, \tag{23}$$





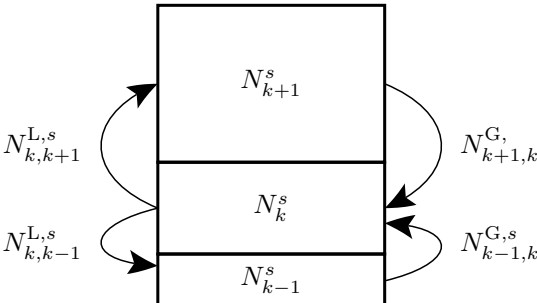

**Figure 2.** Schematic of Eq. (22) depicting number concentrations lost by grid cell $k$ to neighboring cells and the number concentrations gained by grid cell $k$.

where $V_k^s$ is the computational volume of grid cell $k$ at time $t^s$. Applying Eq. (23) to Eq. (22) and multiplying by $V_k^s$ yields the following equation for the real-valued particle number

$$
C_k^{s+1} = C_k^s + \underbrace{\underbrace{\left( \frac{\Delta t}{\Delta z_k^s \Delta z_{k+\frac{1}{2}}^s} \frac{\rho_{k+\frac{1}{2}}^s}{\rho_{k+1}^s} K_{\mathrm{h},k+\frac{1}{2}}^s \right)}_{p_{k+1,k}^{\mathrm{G},s}} \frac{V_k^s}{V_{k+1}^s} C_{k+1}^s}_{C_{k+1,k}^{\mathrm{G},s}} - \underbrace{\underbrace{\left( \frac{\Delta t}{\Delta z_k^s \Delta z_{k+\frac{1}{2}}^s} \frac{\rho_{k+\frac{1}{2}}^s}{\rho_k^s} K_{\mathrm{h},k+\frac{1}{2}}^s \right)}_{p_{k,k+1}^{\mathrm{L},s}} C_k^s}_{C_{k,k+1}^{\mathrm{L},s}}
$$

$$
- \underbrace{\underbrace{\left( \frac{\Delta t}{\Delta z_k^s \Delta z_{k-\frac{1}{2}}^s} \frac{\rho_{k-\frac{1}{2}}^s}{\rho_k^s} K_{\mathrm{h},k-\frac{1}{2}}^s \right)}_{p_{k,k-1}^{\mathrm{L},s}} C_k^s}_{C_{k,k-1}^{\mathrm{L},s}} + \underbrace{\underbrace{\left( \frac{\Delta t}{\Delta z_k^s \Delta z_{k-\frac{1}{2}}^s} \frac{\rho_{k-\frac{1}{2}}^s}{\rho_{k-1}^s} K_{\mathrm{h},k-\frac{1}{2}}^s \right)}_{p_{k-1,k}^{\mathrm{G},s}} \frac{V_k^s}{V_{k-1}^s} C_{k-1}^s}_{C_{k-1,k}^{\mathrm{G},s}}, \tag{24}
$$

where naming conventions are used similarly to Eq. (22) with particle number $C$ replacing number concentration $N$ and coefficient $p$ replacing $\beta$.

### 3.1.2 Stochastic turbulent transport of particles

Equation (24) expresses the gains and losses of grid cell $k$ in terms of deterministic real-valued particle number $C_k$. However, the PartMC model simulates a finite set of particles for each grid cell rather than a number concentration or deterministic real-valued particle number. Therefore, an additional step is required to transform equations from the deterministic real-valued particle number to an integer number of particles that are lost and gained from the finite particle population $\Pi_k$. The set of particles for grid cell $k$ progresses from time $s$ to $s+1$ by

$$
\Pi_k^{s+1} = \Pi_k^s \uplus \Pi_{k+1,k}^{\mathrm{G},s} \setminus \Pi_{k,k+1}^{\mathrm{L},s} \setminus \Pi_{k,k-1}^{\mathrm{L},s} \uplus \Pi_{k-1,k}^{\mathrm{G},s}, \tag{25}
$$

where each $\Pi$ is a finite set of particles, $\uplus$ is the multiset sum, and $\setminus$ is the multiset difference. We use the same subscript and superscript notation as previously.





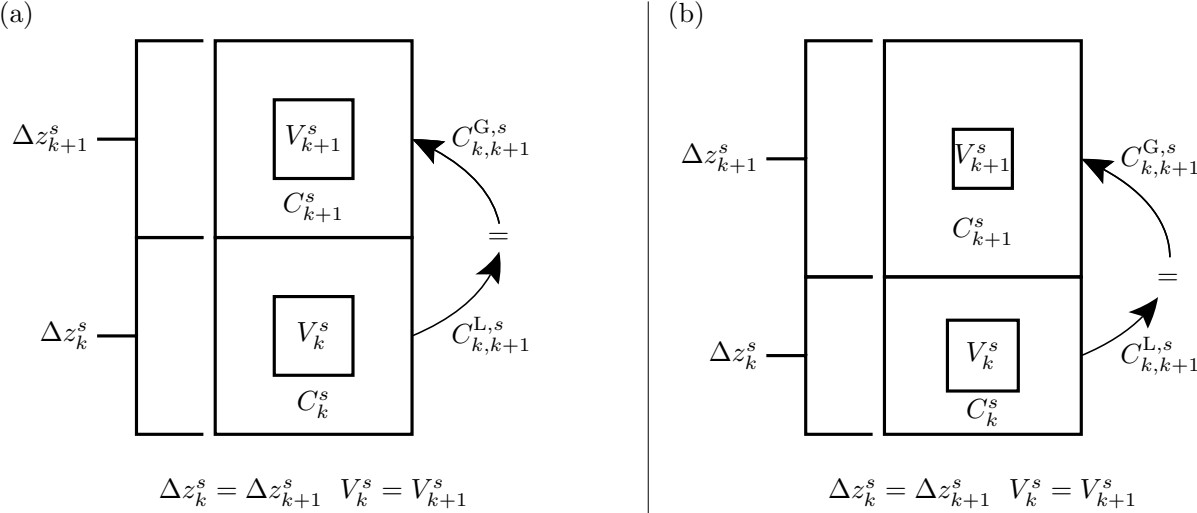

**Figure 3.** Schematic of the transport process for particle number from grid cell $k$ to $k+1$. Panel (a) shows the scenario when the particle number removed from grid cell $k$ is equal to the particle number added to grid cell $k+1$. Panel (b) shows the scenario when the particle number lost from grid cell $k$ is not equal to the particle number gained by $k+1$, which arises when the gain and loss probabilities are unequal due to differences in grid cell sizes and computational volumes.

To determine the gain and loss sets in Eq. (25) we discretize the deterministic real-valued particle number gains and losses of Eq. (24) by applying binomial sampling of the particle set, where a binomial sample of the finite particle set $\Pi$ with a probability $p$ of selecting each particle is denoted by $\mathrm{Binom}(\Pi, p)$ for $0 \le p \le 1$. For example, the discretization of real-valued particle number from cell $k+1$ to $k$, $C_{k+1,k}^{\mathrm{G},s}$, with coefficient $p_{k+1,k}^{\mathrm{G},s}$ is given by

$$\Pi_{k+1,k}^{\mathrm{G},s} \sim \mathrm{Binom}(\Pi_{k+1}^{s}, p_{k+1,k}^{\mathrm{G},s}), \tag{26}$$

which is a stochastic discretization of the corresponding deterministic equation $C_{k+1,k}^{\mathrm{G},s} = p_{k+1,k}^{\mathrm{G},s} C_{k+1}^{s}$.

From Eq. (24), the particle numbers involved in transport from grid cell $k$ to grid cell $k+1$ are

$$C_{k,k+1}^{\mathrm{G},s} = p_{k,k+1}^{\mathrm{G},s} C_{k}^{s} = \frac{\Delta t}{\Delta z_{k+1}^{s} \Delta z_{k+\frac{1}{2}}^{s}} \frac{\rho_{k+\frac{1}{2}}^{s}}{\rho_{k}^{s}} K_{\mathrm{h},k+\frac{1}{2}}^{s} \frac{V_{k+1}^{s}}{V_{k}^{s}} C_{k}^{s}, \tag{27}$$

$$C_{k,k+1}^{\mathrm{L},s} = p_{k,k+1}^{\mathrm{L},s} C_{k}^{s} = \frac{\Delta t}{\Delta z_{k}^{s} \Delta z_{k+\frac{1}{2}}^{s}} \frac{\rho_{k+\frac{1}{2}}^{s}}{\rho_{k}^{s}} K_{\mathrm{h},k+\frac{1}{2}}^{s} C_{k}^{s}. \tag{28}$$

Comparing Eq. (27) and Eq. (28), we see that these gain and loss numbers are generally different from one another due to two factors, namely the difference in the vertical grid cell sizes and the difference in the computational volumes containing the particles as illustrated in Fig. 3.

When we sample the sets of gain and loss populations, we wish to ensure that the gain and loss populations have as many particles in common as possible. This is done in order to minimize particle duplications and removals. To accomplish this, it is





convenient to define the ratio of loss to gain by

$$\gamma_{k,k+1}^s = \frac{C_{k,k+1}^{\mathrm{L},s}}{C_{k,k+1}^{\mathrm{G},s}} = \frac{p_{k,k+1}^{\mathrm{L},s}}{p_{k,k+1}^{\mathrm{G},s}} = \frac{\Delta z_{k+1}^s}{\Delta z_k^s} \frac{V_k^s}{V_{k+1}^s}. \tag{29}$$

This ratio contains the quantities that cause the particle number lost from grid cell $k$ to $k+1$ to be different from the particle number gained by grid cell $k+1$ from $k$. Note that

$$\gamma_{k,k+1}^s = \frac{1}{\gamma_{k+1,k}^s}. \tag{30}$$

To construct particle sets that are as similar as possible, we manipulate the gain and loss terms, $C_{k,k+1}^{\mathrm{G},s}$ and $C_{k,k+1}^{\mathrm{L},s}$, as follows:

$$C_{k,k+1}^{\mathrm{G},s} = p_{k,k+1}^{\mathrm{G},s} C_k^s = \underbrace{\max(p_{k,k+1}^{\mathrm{L},s}, p_{k,k+1}^{\mathrm{G},s})}_{p_{k,k+1}^{\mathrm{T},s}} \min(1/p_{k,k+1}^{\mathrm{L},s}, 1/p_{k,k+1}^{\mathrm{G},s}) p_{k,k+1}^{\mathrm{G},s} C_k^s, \tag{31}$$

$$= C_{k,k+1}^{\mathrm{T},s} \min(1/\gamma_{k,k+1}^s, 1), \tag{32}$$

$$C_{k,k+1}^{\mathrm{L},s} = p_{k,k+1}^{\mathrm{L},s} C_k^s = \underbrace{\max(p_{k,k+1}^{\mathrm{L},s}, p_{k,k+1}^{\mathrm{G},s})}_{p_{k,k+1}^{\mathrm{T},s}} \min(1/p_{k,k+1}^{\mathrm{L},s}, 1/p_{k,k+1}^{\mathrm{G},s}) p_{k,k+1}^{\mathrm{L},s} C_k^s \tag{33}$$

$$= C_{k,k+1}^{\mathrm{T},s} \min(1, \gamma_{k,k+1}^s), \tag{34}$$

where the maximum transport term $C_{k,k+1}^{\mathrm{T},s}$ and transport probability $p_{k,k+1}^{\mathrm{T},s}$ are defined as

$$C_{k,k+1}^{\mathrm{T},s} = p_{k,k+1}^{\mathrm{T},s} C_k^s \tag{35}$$

$$p_{k,k+1}^{\mathrm{T},s} = \max(p_{k,k+1}^{\mathrm{L},s}, p_{k,k+1}^{\mathrm{G},s}). \tag{36}$$

This ensures that $C_{k,k+1}^{\mathrm{G},s}$ and $C_{k,k+1}^{\mathrm{L},s}$ are always less than or equal to $C_{k,k+1}^{\mathrm{T},s}$, with at least one of them being equal. Applying binomial sampling to Eq. (35), we obtain the finite transport particle set

$$\Pi_{k,k+1}^{\mathrm{T},s} \sim \mathrm{Binom}\Big(\Pi_k^s, p_{k,k+1}^{\mathrm{T},s}\Big). \tag{37}$$

Here $\Pi_{k,k+1}^{\mathrm{T},s}$ is the set of particles which are candidates for addition to $k+1$ and removal from $\Pi_k$. We then determine the gain and loss particle sets from $\Pi_{k,k+1}^{\mathrm{T},s}$ as

$$\Pi_{k,k+1}^{\mathrm{G},s} \sim \mathrm{Binom}\Big(\Pi_{k,k+1}^{\mathrm{T},s}, \min(1, 1/\gamma_{k,k+1}^s)\Big) \tag{38}$$

$$\Pi_{k,k+1}^{\mathrm{L},s} \sim \mathrm{Binom}\Big(\Pi_{k,k+1}^{\mathrm{T},s}, \min(1, \gamma_{k,k+1}^s)\Big). \tag{39}$$

As a result of this approach, the particles in the larger of the particle sets contains all the particles that are in the transfer set and the smaller of the particle sets is a subset of those. Binomial samples satisfy a conditional property. If $X \sim \mathrm{Binom}(N, p)$ and, conditional on $X$, $Y \sim \mathrm{Binom}(X, q)$, then $Y \sim \mathrm{Binom}(N, pq)$. Using this property, Eq. (38) and Eq. (39) can be combined with Eq. (37) to give

$$\Pi_{k,k+1}^{\mathrm{G},s} \sim \mathrm{Binom}\Big(\Pi_k^s, p_{k,k+1}^{\mathrm{G},s}\Big) \tag{40}$$

$$\Pi_{k,k+1}^{\mathrm{L},s} \sim \mathrm{Binom}\Big(\Pi_k^s, p_{k,k+1}^{\mathrm{L},s}\Big). \tag{41}$$





However, $\Pi_{k,k+1}^{\mathrm{G},s}$ and $\Pi_{k,k+1}^{\mathrm{L},s}$ are not independent and so they must be sampled via Eqs. (37)-(39) and not Eqs. (40)-(41).

### 3.1.3 Stochastic particle transport algorithm

In the previous section we discussed how particles are transported between a pair of grid cells. However, turbulent mixing involves interactions between grid cell $k$ and both neighboring grid cells, $k+1$ and $k-1$. All equations from the previous

section can be extended to determine the interactions between these grid cells. We can sample particles transported in both directions from particle population $\Pi_k^s$ as

$$\Pi_{k,k+1}^{\mathrm{T},s}, \Pi_{k,k-1}^{\mathrm{T},s}, \Pi_k^{\mathrm{U},s} \sim \mathrm{Mult}\left(\Pi_k^s, p_{k,k+1}^{\mathrm{T},s}, p_{k,k-1}^{\mathrm{T},s}, 1 - p_{k,k+1}^{\mathrm{T},s} - p_{k,k-1}^{\mathrm{T},s}\right), \tag{42}$$

where $\mathrm{Mult}\left(\Pi, p_1, p_2, p_3\right)$ is a multinomial distribution that samples particles into three subpopulations according to the three probabilities, which must sum to one: $p_1 + p_2 + p_3 = 1$. In practice, this is evaluated as the equivalent series of binomials

$$\Pi_{k,k+1}^{\mathrm{T},s} \sim \mathrm{Binom}\left(\Pi_k^s, p_{k,k+1}^{\mathrm{T},s}\right) \tag{43}$$

$$\Pi_{k,k-1}^{\mathrm{T},s} \sim \mathrm{Binom}\left(\Pi_k^s \setminus \Pi_{k,k+1}^{\mathrm{T},s}, \frac{p_{k,k-1}^{\mathrm{T},s}}{1 - p_{k,k+1}^{\mathrm{T},s}}\right) \tag{44}$$

$$\Pi_k^{\mathrm{U},s} = \Pi_k^s \setminus \Pi_{k,k+1}^{\mathrm{T},s} \setminus \Pi_{k,k-1}^{\mathrm{T},s}, \tag{45}$$

where $\Pi_k^{\mathrm{U},s}$ is the set of unsampled particles in $k$. The gain and loss sets are then given by

$$\Pi_{k,k+1}^{\mathrm{G},s} \sim \mathrm{Binom}\left(\Pi_{k,k+1}^{\mathrm{T},s}, \min\left(1, 1/\gamma_{k,k+1}\right)\right) \tag{46}$$

$$\Pi_{k,k+1}^{\mathrm{L},s} \sim \mathrm{Binom}\left(\Pi_{k,k+1}^{\mathrm{T},s}, \min\left(1, \gamma_{k,k+1}\right)\right) \tag{47}$$

$$\Pi_{k,k-1}^{\mathrm{G},s} \sim \mathrm{Binom}\left(\Pi_{k,k-1}^{\mathrm{T},s}, \min\left(1, 1/\gamma_{k,k-1}\right)\right) \tag{48}$$

$$\Pi_{k,k-1}^{\mathrm{L},s} \sim \mathrm{Binom}\left(\Pi_{k,k-1}^{\mathrm{T},s}, \min\left(1, \gamma_{k,k-1}\right)\right). \tag{49}$$

Some of the particles initially sampled into the transport sets $\Pi_{k,k+1}^{\mathrm{T},s}$ and $\Pi_{k,k-1}^{\mathrm{T},s}$ are not lost and will remain in cell $k$:

$$\Pi_{k,k+1}^{\mathrm{R},s} = \left(\Pi_{k,k+1}^{\mathrm{T},s} \setminus \Pi_{k,k+1}^{\mathrm{L},s}\right) \tag{50}$$

$$\Pi_{k,k-1}^{\mathrm{R},s} = \left(\Pi_{k,k-1}^{\mathrm{T},s} \setminus \Pi_{k,k-1}^{\mathrm{L},s}\right). \tag{51}$$

Finally, the sets above can be combined to give the new set of particles $\Pi_k^{s+1}$ in cell $k$ by:

$$\Pi_k^{s+1} = \Pi_k^{\mathrm{U},s} \uplus \Pi_{k,k+1}^{\mathrm{R},s} \uplus \Pi_{k,k-1}^{\mathrm{R},s} \uplus \Pi_{k-1,k}^{\mathrm{G},s} \uplus \Pi_{k+1,k}^{\mathrm{G},s}. \tag{52}$$

### 3.1.4 Selection of sub-cycle time step

To maintain numerical stability with the explicit finite volume scheme, sub-cycle time steps are taken for vertical transport that

differ from the model time step for other processes. To determine an appropriate time step, the transfer rates are computed for





all grid cells in the column. Given these values, the critical time step is taken to be

$$\Delta t_{\mathrm{c}} = \frac{1}{2} \max_{k=1,\ldots,n_z} \left( \max \left( \frac{\Delta t}{p_{k,k+1}^{\mathrm{T},s}}, \frac{\Delta t}{p_{k,k-1}^{\mathrm{T},s}} \right) \right). \tag{53}$$

We then determine the number of time steps required to reach the full model time step $\Delta t$ by

$$n_{\mathrm{T}} = \left\lceil \frac{\Delta t}{\Delta t_{\mathrm{c}}} \right\rceil, \tag{54}$$

where $\lceil \cdot \rceil$ is the ceiling function, and define the sub-cycle time step for the transport as

$$\Delta t_{\mathrm{T}} = \frac{\Delta t}{n_{\mathrm{T}}}. \tag{55}$$

This ensures that the sum of the transport probabilities computed for time step $\Delta t_{\mathrm{T}}$ is always in $[0,1]$. That is, $(\Delta t_{\mathrm{T}}/\Delta t)(p_{k,k+1}^{\mathrm{T},s} + p_{k,k-1}^{\mathrm{T},s}) \leq 1$ for all $k$. To see this, we observe that (54) and (55) imply that $\Delta t_{\mathrm{T}} \leq \Delta t_{\mathrm{c}}$. From (53) we have that $\Delta t_{\mathrm{c}} \, p_{k,k+1}^{\mathrm{T},s} \leq \Delta t/2$, and similarly for $p_{k,k-1}^{\mathrm{T},s}$, giving the desired result. Note that scaling the probabilities by the ratio $\Delta t_{\mathrm{T}}/\Delta t$ is the same as computing the probabilities with time step $\Delta t_{\mathrm{T}}$ because the probabilities are linear in $\Delta t$.

The complete algorithm for stochastic turbulent transport of finite particle sets is given in Algorithms 1 and 2. The $\mathrm{Binom}(\Pi, p)$ binomial samples reflect the fact each particle has equal probability of being transported. A $\mathrm{Binom}(\Pi, p)$ sample can be implemented by first sampling a scalar binomial function $n \sim \mathrm{Binom}(N_{\mathrm{p}}, p)$ where $N_{\mathrm{p}} = |\Pi|$ is the number of particles in population $\Pi$, and then choosing $n$ particles uniformly from $\Pi$.

### 3.1.5 Rebalancing of computational particle number

During a given simulation, the number of computational particles changes as particles are added due to emission, are transferred from one grid cell to another due to turbulent transport, and are removed by coagulation and dry deposition. When the number of computational particles falls below half the initial prescribed number in a given grid cell, in order to maintain an adequate statistical sample, we duplicate every particle and double the computational volume. When the number of particles is twice the initially prescribed number, in order to alleviate the higher computational cost, half the computational particles are discarded and the computational volume is halved. This strategy has been previously used for particle populations in the 0-D box model PartMC-MOSAIC (Riemer et al., 2009) as well as other Monte Carlo simulations for particle dynamics (Efendiev and Zachariah, 2002; Maisels et al., 2004).

### 3.2 Aerosol dry deposition algorithm

Particles near the surface are subject to removal by the process of dry deposition. This is parameterized by evaluating a dry deposition velocity for each particle in the aerosol population of the lowest grid cell $\Pi_1$. The parameterization presented in Zhang et al. (2001) is applied here on a per-particle basis. The dry deposition velocity is dependent on the per-particle diameter and density, in addition to the meteorological conditions and surface characteristics. Given a dry deposition velocity for particle $i$, denoted $V_{\mathrm{d},i}$, a loss rate probability, $\ell_{\mathrm{d},i}$, can be expressed as

$$\ell_{\mathrm{d},i} = \frac{\Delta t}{\Delta z_1} V_{\mathrm{d},i}, \tag{56}$$





---

**Algorithm 1** Stochastic aerosol particle 1D transport method for one timestep with sub-cycling.

---

**function** $\left( \{\Pi_k^0\}_{k=1}^{n_z}, \{V_k\}_{k=1}^{n_z}, \Delta t, \{\Delta z_k\}_{k=1}^{n_z}, \{\Delta z_{k+1/2}\}_{k=1}^{n_z-1}, \{\rho_k\}_{k=1}^{n_z}, \{\rho_{k+1/2}\}_{k=1}^{n_z-1}, \{K_{k+1/2}\}_{k=1}^{n_z-1} \right) \to \left( \{\Pi_k\}_{k=1}^{n_z} \right)$

**Input:**

$\{\Pi_k^0\}_{k=1}^{n_z}$ are the grid-cell particle populations at the start of the time step

$\{V_k\}_{k=1}^{n_z}$ are the computational volumes for the particle populations

$\Delta t$ is the time step

$\{\Delta z_k\}_{k=1}^{n_z}$ and $\{\Delta z_{k+1/2}\}_{k=1}^{n_z-1}$ are the grid dimensions (see Figure 1)

$\{\rho_k\}_{k=1}^{n_z}$ and $\{\rho_{k+1/2}\}_{k=1}^{n_z-1}$ are the air densities at grid cell centers and boundaries, respectively

$\{K_{k+1/2}\}_{k=1}^{n_z-1}$ are the diffusion coefficients at grid cell boundaries

**Output:**

$\Pi_k$ are the updated particle populations for grid cells $k = 1, \ldots n_z$ at the end of the time step

$\tilde{p}_{k,k+1}^{\mathrm{G}} \leftarrow \frac{1}{\Delta z_{k+1}\Delta z_{k+1/2}} \frac{\rho_{k+1/2}}{\rho_k} \frac{V_{k+1}}{V_k} K_{\mathrm{h},k+1/2}$ for $k = 1, \ldots, (n_z - 1)$

$\tilde{p}_{k,k-1}^{\mathrm{G}} \leftarrow \frac{1}{\Delta z_{k-1}\Delta z_{k-1/2}} \frac{\rho_{k-1/2}}{\rho_k} \frac{V_{k-1}}{V_k} K_{\mathrm{h},k-1/2}$ for $k = 2, \ldots, n_z$

$\tilde{p}_{k,k+1}^{\mathrm{L}} \leftarrow \frac{1}{\Delta z_k\Delta z_{k+1/2}} \frac{\rho_{k+1/2}}{\rho_k} K_{\mathrm{h},k+1/2}$ for $k = 1, \ldots, (n_z - 1)$

$\tilde{p}_{k,k-1}^{\mathrm{L}} \leftarrow \frac{1}{\Delta z_k\Delta z_{k-1/2}} \frac{\rho_{k-1/2}}{\rho_k} K_{\mathrm{h},k-1/2}$ for $k = 2, \ldots, n_z$

$\Delta t_{\mathrm{c}} \leftarrow \frac{1}{2} \max_{k=1,\ldots,(n_z-1)} \left( \max \left( \tilde{p}_{k,k+1}^{\mathrm{G}}, \tilde{p}_{k+1,k}^{\mathrm{G}}, \tilde{p}_{k,k+1}^{\mathrm{L}}, \tilde{p}_{k+1,k}^{\mathrm{L}} \right) \right)$

$n_{\mathrm{T}} \leftarrow \left\lceil \frac{\Delta t}{\Delta t_{\mathrm{c}}} \right\rceil$

$\Delta t_{\mathrm{T}} \leftarrow \frac{\Delta t}{n_{\mathrm{T}}}$

$p_{k,k+1}^{\mathrm{G}} \leftarrow \Delta t_{\mathrm{T}} \, \tilde{p}_{k,k+1}^{\mathrm{G}}; \quad p_{k+1,k}^{\mathrm{G}} \leftarrow \Delta t_{\mathrm{T}} \, \tilde{p}_{k+1,k}^{\mathrm{G}}; \quad p_{k,k+1}^{\mathrm{L}} \leftarrow \Delta t_{\mathrm{T}} \, \tilde{p}_{k,k+1}^{\mathrm{L}}; \quad p_{k+1,k}^{\mathrm{L}} \leftarrow \Delta t_{\mathrm{T}} \, \tilde{p}_{k+1,k}^{\mathrm{L}}$ for $k = 1, \ldots, (n_z - 1)$

$\Pi_k \leftarrow \Pi_k^0$ for $k = 1, \ldots, n_z$

**for** $s_{\mathrm{T}} \leftarrow 1$ **to** $n_{\mathrm{T}}$ **do**

    **for** $k \leftarrow 2$ **to** $(n_z - 1)$ **do**

        $p_{\mathrm{sum}} \leftarrow 0$

        $\left( \Pi_{k,k+1}^{\mathrm{G}}, \Pi_{k,k+1}^{\mathrm{R}}, \Pi_k^{\mathrm{U}}, p_{\mathrm{sum}} \right) \leftarrow \mathrm{PARTICLESAMPLE} \left( \Pi_k, p_{k,k+1}^{\mathrm{G}}, p_{k,k+1}^{\mathrm{L}}, p_{\mathrm{sum}} \right)$

        $\left( \Pi_{k,k-1}^{\mathrm{G}}, \Pi_{k,k-1}^{\mathrm{R}}, \Pi_k^{\mathrm{U}}, p_{\mathrm{sum}} \right) \leftarrow \mathrm{PARTICLESAMPLE} \left( \Pi_{k,k+1}^{\mathrm{U}}, p_{k,k-1}^{\mathrm{G}}, p_{k,k-1}^{\mathrm{L}}, p_{\mathrm{sum}} \right)$

    **end for**

    $\left( \Pi_{k,k+1}^{\mathrm{G}}, \Pi_{k,k+1}^{\mathrm{R}}, \Pi_k^{\mathrm{U}}, p_{\mathrm{sum}} \right) \leftarrow \mathrm{PARTICLESAMPLE} \left( \Pi_k, p_{k,k+1}^{\mathrm{G}}, p_{k,k+1}^{\mathrm{L}}, 0 \right)$ for $k = 1$

    $\left( \Pi_{k,k-1}^{\mathrm{G}}, \Pi_{k,k-1}^{\mathrm{R}}, \Pi_k^{\mathrm{U}}, p_{\mathrm{sum}} \right) \leftarrow \mathrm{PARTICLESAMPLE} \left( \Pi_k, p_{k,k-1}^{\mathrm{G}}, p_{k,k-1}^{\mathrm{L}}, 0 \right)$ for $k = n_z$

    $\Pi_{n_z+1,n_z}^{\mathrm{G}} \leftarrow \varnothing; \quad \Pi_{0,1}^{\mathrm{G}} \leftarrow \varnothing$

    **for** $k = 1$ **to** $n_z$ **do**

        $\Pi_k \leftarrow \Pi_k^{\mathrm{U}} \uplus \Pi_{k,k+1}^{\mathrm{R}} \uplus \Pi_{k,k-1}^{\mathrm{R}} \uplus \Pi_{k-1,k}^{\mathrm{G}} \uplus \Pi_{k+1,k}^{\mathrm{G}}$

    **end for**

**end for**

**end function**

---





---

**Algorithm 2** Stochastic aerosol particle transport from a grid cell.

---

**function** PARTICLESAMPLE $\left(\Pi, p^{\mathrm{G}}, p^{\mathrm{L}}, p_{\mathrm{prev}}\right) \rightarrow \left(\Pi^{\mathrm{G}}, \Pi^{\mathrm{R}}, \Pi^{\mathrm{U}}, p_{\mathrm{new}}\right)$

**Input:**

$\Pi$ is the starting particle population of the source grid cell

$p^{\mathrm{G}}$ is the particle gain probability

$p^{\mathrm{L}}$ is the particle loss probability

$p_{\mathrm{prev}}$ is the sum of previous transfer probabilities

**Output:**

$\Pi^{\mathrm{G}}$ is the particle population to be gained by the destination grid cell

$\Pi^{\mathrm{R}}$ is the particle population that is temporarily removed from the source grid cell

$\Pi^{\mathrm{U}}$ is the unsampled particle population in the source grid cell

$p_{\mathrm{new}}$ is the updated sum of transfer probabilities

$p^{\mathrm{T}} = \frac{1}{1-p_{\mathrm{prev}}} \max\left(p^{\mathrm{L}}, p^{\mathrm{G}}\right)$

$\Pi^{T} \sim \mathrm{Binom}\left(\Pi^{\mathrm{T}}, p^{\mathrm{T}}\right)$

$\Pi^{\mathrm{U}} \leftarrow \Pi \setminus \Pi^{\mathrm{T}}$

$\gamma \leftarrow p^{\mathrm{L}}/p^{\mathrm{G}}$

**if** $\gamma \geq 1$ **then**

   $\Pi^{\mathrm{G}} \sim \mathrm{Binom}\left(\Pi^{\mathrm{T}}, \frac{1}{\gamma}\right)$

   $\Pi^{\mathrm{R}} \leftarrow \varnothing$

**else**

   $\Pi^{\mathrm{G}} \leftarrow \Pi^{\mathrm{T}}$

   $\Pi^{\mathrm{L}} \sim \mathrm{Binom}\left(\Pi^{\mathrm{T}}, \gamma\right)$

   $\Pi^{\mathrm{R}} \leftarrow \Pi^{\mathrm{T}} \setminus \Pi^{\mathrm{L}}$

**end if**

$p_{\mathrm{new}} \leftarrow p_{\mathrm{prev}} + p^{\mathrm{T}}$

**end function**

---



where $\Delta z_1$ is the lowest model layer thickness. Algorithm 3 shows the procedure for the removal of particles from the particle population $\Pi_1$, where each particle $i$ is tested for removal with the associated dry deposition loss rate probability $\ell_{\mathrm{d},i}$.

---

**Algorithm 3** Stochastic aerosol particle dry deposition for one time step

---

**function** $\left( \Pi_1^0, \{l_{\mathrm{d},i}\}_{i=1}^{N_{\mathrm{p},1}} \right) \rightarrow (\Pi_1)$

**Input:**

$\Pi_1^0$ is the initial particle population in the grid cell closest to surface at the start of the time step

$\{l_{\mathrm{d},i}\}_{i=1}^{N_{\mathrm{p},1}}$ are the particle loss rate probabilities for each particle in $\Pi_1^0$; see Eq. (56)

**Output:**

$\Pi_1$ is the updated particle population with particle removals due to dry deposition after the time step

$\Pi_1 \leftarrow \Pi_1^0$     (copy the initial particle population)

**for** $i = 1, \ldots, N_{\mathrm{p}}^1$ **do**

   $r \sim [0, 1]$     (randomly sample $r$ in $[0, 1]$)

   **if** $r < \ell_{\mathrm{d},i}$ **then**

      $\Pi_1 \leftarrow \Pi_1 \setminus \{\boldsymbol{\mu}^i\}$     (remove particle $i$)

   **end if**

**end for**

---

## 4   Verification of the stochastic particle algorithms

In the following section, we will present separate numerical verifications of the new algorithms for particle transport by turbulent diffusion (Sec. 4.1) and particle removal by dry deposition (Sec. 4.2).

### 4.1   Particle transport by diffusion

For verification of the stochastic transport method of particles presented in Algorithm 1, the particle transport code was implemented to solve the 1D diffusion equation given by

$$\frac{\partial N(z,t)}{\partial t} = K_{\mathrm{h}} \frac{\partial^2 N(z,t)}{\partial z^2}, \tag{57}$$

where $N(z,t)$ is total aerosol number concentration as defined by Eq. (11). Here, the eddy diffusion coefficient $K_{\mathrm{h}}$ is taken to be constant in both time and space so that the equation can be solved analytically. To isolate diffusion, all other aerosol processes that may contribute to the evolution of the aerosol state were excluded from the simulation. Reflective boundary conditions were imposed at the surface and at the top of the domain. The model was initialized with an instantaneous area source in the x-y plane, with an initial thickness $\Delta z$, centered at altitude $z'$ and with uniform perturbation number concentration $N_0$ and a




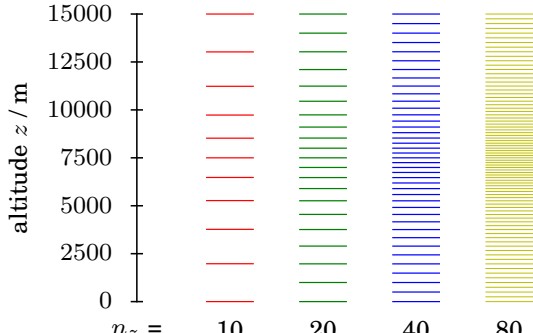

**Figure 4.** Variable grid cell edges for the transport test case domain with 10, 20, 40 and 80 vertical layers.

background number concentration $N_{back}$. The analytical solution for comparison to model results is

$$N(z,t) = N_{back} + \frac{N_0}{2}\left[\mathrm{erf}\left(\frac{0.5\Delta z + (z - z')}{\sqrt{4K_h t}}\right) + \mathrm{erf}\left(\frac{0.5\Delta z - (z - z')}{\sqrt{4K_h t}}\right)\right.$$
$$+ \mathrm{erf}\left(\frac{0.5\Delta z + (z - z'_a)}{\sqrt{4K_h t}}\right) + \mathrm{erf}\left(\frac{0.5\Delta z - (z - z'_a)}{\sqrt{4K_h t}}\right)$$
$$\left. + \mathrm{erf}\left(\frac{0.5\Delta z + (z - z'_b)}{\sqrt{4K_h t}}\right) + \mathrm{erf}\left(\frac{0.5\Delta z - (z - z'_b)}{\sqrt{4K_h t}}\right)\right], \tag{58}$$

where the method of images was applied to impose boundary conditions, with imaginary sources at $z'_a$ and $z'_b$. In this diffusion
test case the background particle concentration was $N_{back} = 3.2 \times 10^3 \mathrm{\ cm^{-3}}$. The instantaneous finite-thickness particle cloud
was placed at the altitude $z' = 7500 \mathrm{\ m}$ with thickness $\Delta z = 2000 \mathrm{\ m}$ and consisted of a perturbation particle number concen-
tration of $N_0 = 3.2 \times 10^4 \mathrm{\ cm^{-3}}$. Other input parameters were $z = 0 \mathrm{\ m}$ and $z = 15\,000 \mathrm{\ m}$ for the altitudes of the surface and
the top of the model domain, respectively. The altitudes for the imaginary sources were $z'_a = -7500 \mathrm{\ m}$ and $z'_b = 22500 \mathrm{\ m}$, and
the turbulent diffusion coefficient was $K_h = 50 \mathrm{\ m^2\ s^{-1}}$.
The variable grid spacing was determined by

$$\Delta z_k = C_0\left(3 + \cos\left(\frac{2\pi(k - \frac{1}{2})}{n_z}\right)\right), \tag{59}$$

with $C_0$ as a domain scaling parameter given by $C_0 = 15000/(3n_z)$. The grid cell edges are determined by

$$z_{k+\frac{1}{2}} = \sum_{i=1}^{k} C_0\left(3 + \cos\left(\frac{2\pi(k - \frac{1}{2})}{n_z}\right)\right), \tag{60}$$

for $k = 1, \ldots, n_z$.
Figure 4 shows the variation in vertical spacing of grid cells resulting from Eq. (60) where the small grid cells, located at
7500 m, are approximately half as large as the largest grid cells, located at the domain edges. We use a sequence of grids, each
with twice as many grid cells as the last, giving $n_z = 10, 20, 40, 80$.

Figure 5 shows the aerosol particle number concentration evolution for a single simulation with $n_z = 20$ grid cells, using
$N_p = 10^4$ computational particles. The simulated number concentration is compared to the analytical Gaussian solution and





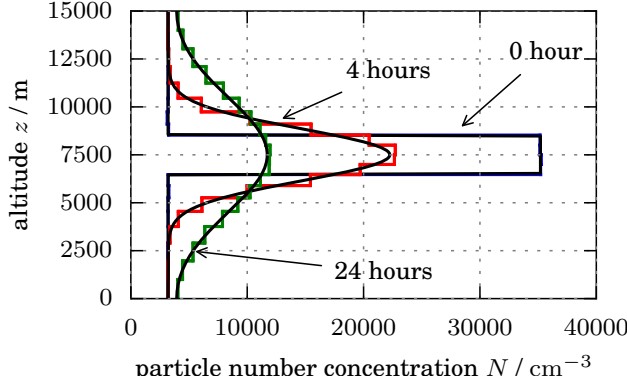

**Figure 5.** Number concentration as predicted by the model at $t = 0$ h (blue), $t = 4$ h (red), and $t = 24$ h (green) with analytical solution (black) for an instantaneous area source released in the center of the domain at $t = 0$.

shows good agreement. The noise in the number concentration is a result of stochastic sampling and could be further reduced by averaging several independent simulations to form an ensemble mean.

To verify the convergence of the transport algorithm to the analytical solution, we quantified the error in the total number concentration for ensemble member $j$ by the weighted $L^2$ error:

$$5 \quad ||\hat{N}^{t,j} - \bar{N}^t||_2 = \sqrt{\sum_{k=1}^{n_z} \left( \hat{N}_k^{t,j} - \bar{N}_k^t \right)^2 \Delta z_k}, \tag{61}$$

where $n_z$ is the total number of grid cells, $\hat{N}_k^{t,j}$ is the stochastic number concentration for grid cell $k$ at time $t$ for ensemble member $j$, $\bar{N}_k^t$ is the average analytical number concentration over grid cell $k$, and $\Delta z_k$ is the grid cell size of cell $k$. The average analytical number concentration of grid cell $k$ is given by

$$\bar{N}_k^t = \frac{1}{\Delta z_k} \int\limits_{z_{k-1/2}^s}^{z_{k+1/2}^s} N(z,t) dz, \tag{62}$$

10     where $N(z,t)$ is the number concentration given by the analytical solution Eq. (58).

The total relative error for ensemble member $j$ is given by

$$e^{t,j} = \frac{||\hat{N}^{t,j} - \bar{N}^t||_2}{||\bar{N}^t||_2}. \tag{63}$$

The ensemble mean error $\bar{e}^t$ is given by

$$\bar{e}^t = \sqrt{\frac{1}{n_{\mathrm{run}}} \sum_{j=1}^{n_{\mathrm{run}}} (e^{t,j})^2} \tag{64}$$



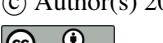

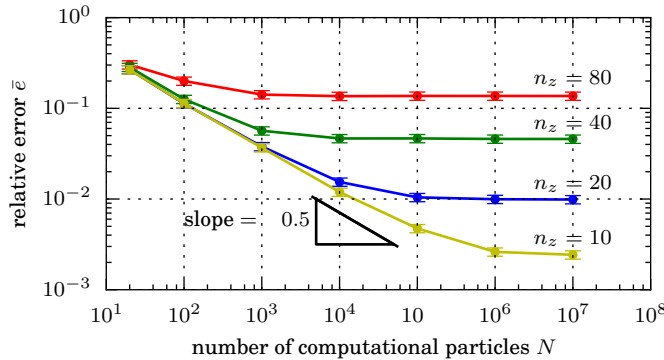
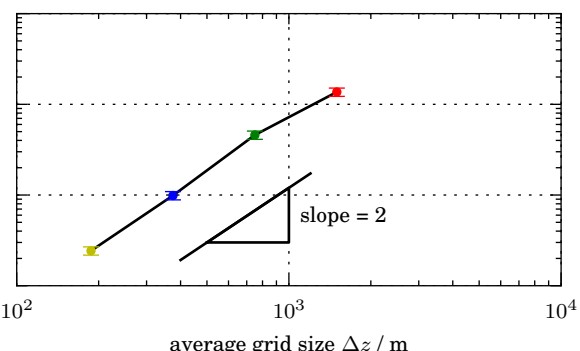

**Figure 6. (a)** Convergence of the stochastic solution to the analytical solution of the 1D diffusion equation. **(b)** Convergence of the error as a function of $\Delta z$ for number of computational particles $N_{\mathrm{p}} \to \infty$. Black triangles show the expected convergence rates.

and the standard deviation in the error $\sigma_e$ given by

$$\sigma_e^t = \sqrt{\frac{1}{n_{\mathrm{run}}} \sum_{j=1}^{n_{\mathrm{run}}} \left(e^{t,j} - \bar{e}^t\right)^2}. \tag{65}$$

Figure 6 shows the error convergence behavior as the number of computational particles, $N_{\mathrm{p}}$, and average grid cell size, $\Delta z$, vary. The average grid cell size is given by the domain height divided by the number of grid cells, so $\Delta z = (15\,000 \text{ m})/n_z$.

The ensemble size was $n_{\mathrm{run}} = 20$ and the number of computational particles $N_p$ ranged from 20 to $10^7$. Error bars represent the 95% confidence interval.

We expect that the stochastic particle solution $\hat{N}_k^t$ will converge to the finite volume solution $N_k^t$ (see Eq. (20)) as $N_{\mathrm{p}} \to \infty$. In turn, we expect $N_k^t$ to converge to the analytical solution $\bar{N}_k^t$ as $\Delta z \to 0$. Thus, we anticipate the convergence

$$\bar{N}_k^t = \lim_{\Delta z \to 0} \lim_{N_{\mathrm{p}} \to \infty} \hat{N}_k^t. \tag{66}$$

To understand the rates of convergence, we decompose the error as

$$\left\| \hat{N}^t - \bar{N}^t \right\|_2 = \left( \sum_{k=1}^{n_z} \Delta z_k \left( \hat{N}_k^t - \bar{N}_k^t \right)^2 \right)^{\frac{1}{2}} \tag{67}$$

$$= \left( \sum_{k=1}^{n_z} \Delta z_k \left( \hat{N}_k^t - N_k^t + N_k^t - \bar{N}_k^t \right)^2 \right)^{\frac{1}{2}}$$

$$\leq \underbrace{\left\| \hat{N}^t - N^t \right\|_2}_{\substack{O\left(1/\sqrt{N_{\mathrm{p}}}\right) \\ \text{stochastic error} \to 0 \\ \text{as } N_{\mathrm{p}} \to \infty}} + \underbrace{\left\| N^t - \bar{N}^t \right\|_2}_{\substack{O\left(\Delta z^2\right) \\ \text{finite volume error} \to 0 \\ \text{as } \Delta z \to 0}}. \tag{68}$$

Here we see that the total error is bounded by the stochastic and finite volume errors. The stochastic error is $O(1/\sqrt{N_{\mathrm{p}}})$, while

the finite volume error is $O(\Delta z^2)$ since the spatial discretization is second order accurate.





Figure 6(a) shows convergence of total error for fixed $\Delta z$ as $N_\mathrm{p} \rightarrow \infty$, with the expected $-1/2$ slope until the finite volume error dominates. Figure 6(b) shows the convergence as the grid size $\Delta z$ decreases for large $N_\mathrm{p}$. Each data point in Fig. 6(b) corresponds to the converged value of a line in Fig. 6(a), taken with $N_\mathrm{p} = 10^7$. In this figure we see the expected slope of 2.

### 4.2 Particle dry deposition

To verify Algorithm 3 for dry deposition we developed a test case that only considered the removal of particles by dry deposition. The simulation was initialized with two monodisperse particle populations with different diameters, 1 μm and 10 μm, and identical densities of $1800 \, \mathrm{kg \, m^{-3}}$. The number concentration of the 1 μm particle population was $10^3$ higher than the number of concentration of the 10 μm population, so that the mass concentrations of the initial particle populations were equal. The dry deposition velocities of the two populations did not change over time due to the absence of coagulation and condensation. Each population was expected to decay at a rate based on their computed dry deposition velocities. As a result, the deposition process can be represented as a first order decay equation

$$\frac{dM_D}{dt} = - \underbrace{\frac{V_{\mathrm{d},D}}{\Delta z_{\mathrm{ref}}}}_{\lambda_D} M_D, \tag{69}$$

where $M_D$ is the aerosol mass concentration of a given population of particles with diameter $D$. The loss rate of the particles, $\lambda_D$, for population with diameter $D$ is given by the deposition velocity of particles in that population, $V_{\mathrm{d},D}$, and the reference height $\Delta z_{\mathrm{ref}}$. The analytical solution is

$$M_D(t) = M_{D,0} \exp\left(-\lambda_D t\right), \tag{70}$$

where $M_{D,0}$ is the initial aerosol mass concentration and $M_D(t)$ is the aerosol mass concentration at time $t$.

Figure 7 shows the evolution of initial identical mass concentrations for two particle populations, one with particles with diameter of 1 μm and the other with diameter of 10 μm. The results are an average of 10 independent model runs. The average mass concentration and 95% confidence interval are shown to be in good agreement with the analytical concentrations as given by Eq. (70). The mass concentration associated with particles with diameter of 10 μm decays more quickly because 10 μm particles have a higher settling velocity than 1 μm particles. Particles with diameter of 1 μm experience a very slow loss in mass concentration as a result of ineffective removal by any of the processes represented by dry deposition. These particles are too large to be removed effectively by Brownian diffusion and too small to be removed by gravitational settling.

## 5 Application of single column model with an idealized scenario

### 5.1 Setup of idealized scenario

We constructed an idealized scenario to illustrate the model capabilities of WRF-PartMC-MOSAIC-SCM. The scenario is similar to the box model study presented in Riemer et al. (2009) and Zaveri et al. (2010), but for the first time we now gain insight into aerosol mixing state as it varies spatially with altitude.



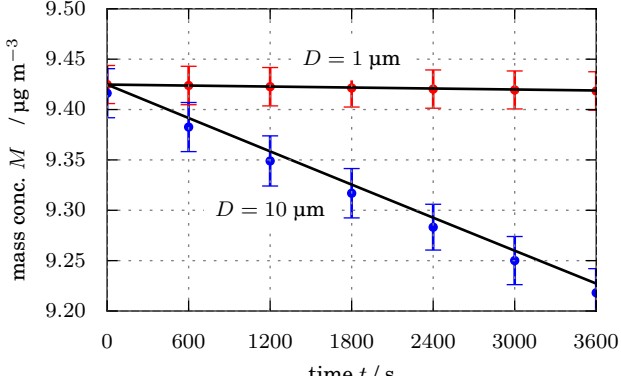

**Figure 7.** Evolution of mass concentration due to dry deposition for particles of diameter 1 μm, indicated by red dots, and particles of diameter 10 μm, indicated by blue dots. Respective analytical results from Eq. (70) are shown as solid lines. The error bars at each point represent 95% confidence intervals from 10 ensemble runs.

We simulated a 48-hour episode, starting at 06:00 local standard time (LST). Initial gas mixing ratios were based on initial conditions given by Riemer et al. (2009) and decreased linearly with height to a height of 3.5 km. Gas phase emissions were specified only at the surface and were also based on the urban plume case described in Riemer et al. (2009), adapted from the Southern California Air Quality Study (SCAQS) simulation (26–29 August 1988 period) of Zaveri et al. (2008). Table 1 shows
the initial aerosol distributions and aerosol emissions used in this scenario with two aerosol modes, an Aitken mode and an accumulation mode. Both aerosol modes consisted of particles that contained ammonium sulfate and primary organic aerosol. Initial aerosol number concentration was constant with height.

Carbonaceous aerosols were emitted at the surface from three different sources: diesel vehicles, gasoline vehicles, and meat cooking. Due to the importance of the timing of atmospheric turbulent mixing and emissions, we applied a diurnal cycle to the
particle emission rates. This is in contrast to Riemer et al. (2009), where the particle emission rates were held constant with time. The gasoline and diesel emission source strengths were varied over time by redistributing the mean aerosol emissions from Riemer et al. (2009) based on the weekday traffic distribution fractions as described in Marr et al. (2002). The resulting 48-hour time series used for this scenario are shown in Fig. 8.

WRF-PartMC-MOSAIC-SCM was initialized with 59 vertical levels, logarithmically spaced with 16 levels in the lowest
1 km of the domain. The Mellor-Yamada-Janjic (MYJ) planetary boundary layer scheme (Janjic, 1994) was used to model turbulent transport and parameterize the diffusion coefficient for the particle transport scheme. The model was initialized with 25 000 computational particles for each level, resulting in a total of nearly 1.5 million particles in the column. The number of particles per layer over the course of the simulation was restricted to a range between half and double the initial number of particles (12 500 and 50 000) to maintain accuracy while avoiding higher computational costs as described in Sec. 3.1.3.





**Table 1.** Initial and emitted aerosol distribution parameters.

| Initial | $N$ / m$^{-3}$ | $D_\text{gn}$ / μm | $\sigma_\text{g}$ | Composition by mass |
|---|---|---|---|---|
| Aitken Mode | $3.2 \times 10^9$ | 0.02 | 1.45 | 50% (NH$_4$)$_2$SO$_4$, 50% POA |
| Accumulation Mode | $2.9 \times 10^9$ | 0.116 | 1.65 | 50% (NH$_4$)$_2$SO$_4$, 50% POA |
| Emissions[a] | $E$ / m$^{-2}$ s$^{-1}$ | $D_\text{gn}$ / μm | $\sigma_\text{g}$ | Composition by mass |
| Meat cooking | $9.0 \times 10^6$ | 0.086 | 1.9 | 100% POA |
| Diesel vehicles | $1.1 \times 10^8$ | 0.05 | 1.7 | 30% POA, 70% BC |
| Gasoline vehicles | $3.5 \times 10^7$ | 0.05 | 1.7 | 80% POA, 20% BC |

[a] Diesel and gasoline vehicle emission values are averaged over the 48-h simulation period.

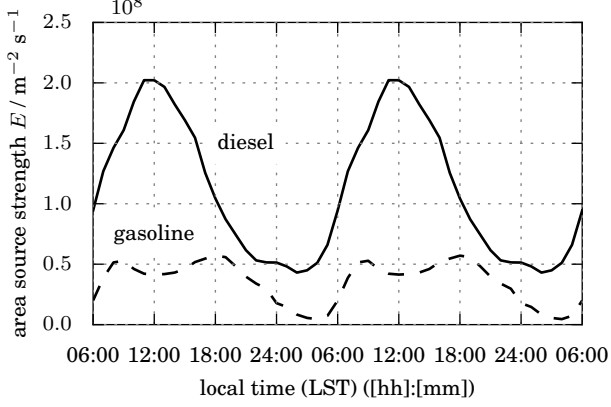

**Figure 8.** Time series of diesel and gasoline area source surface emissions for the 48-hour simulated period.

## 5.2 Aerosol distribution functions

Given the complexities of the multidimensional aerosol distribution, we must project the distribution for purposes of displaying results. We take $N(D)$ to be the cumulative number distribution where $N(D)$ is the number of particles per volume that have a diameter less than $D$. Given the cumulative number distribution, we define the number distribution $n(D)$ as

$$n(D) = \frac{dN(D)}{d\log_{10} D}.$$
(71)

To characterize the aerosol mixing state, we define the per-particle mass fraction of species $a$ as

$$w_{a,\text{dry}} = \frac{\mu_a}{\mu_\text{dry}},$$
(72)

(c) Author(s) 2017. CC BY 3.0 License.





where $\mu_a$ is the mass of species $a$ in a given particle and $\mu_{dry}$ is the total dry mass of the particle. Here species $a$ can be a single aerosol species such as black carbon (BC), sulfate (SO$_4$), or nitrate (NO$_3$), or it consist of a group aerosol species such as secondary organic aerosol (SOA).

The one-dimensional cumulative number distribution $N(w_{a,dry})$ is the number concentration of particles with dry mass fraction of species $a$ less than $w_{a,dry}$. The corresponding number distribution $n(w_{a,dry})$ is defined as

$$n(w_{a,dry}) = \frac{\partial N(w_{a,dry})}{\partial w_{a,dry}}. \tag{73}$$

We can also construct two-dimensional number distributions with respect to dry diameter, $D_{dry}$, and dry mass fraction of species $a$, $w_{a,dry}$. The two-dimensional number distribution $n(D_{dry}, w_{a,dry})$ is defined as

$$n(D_{dry}, w_{a,dry}) = \frac{\partial^2 N(D_{dry}, w_a)}{\partial \log_{10} D_{dry} \partial w_a}, \tag{74}$$

where $N(D_{dry}, w_a)$ is the number concentration of particles that have dry diameter less than $D_{dry}$ and dry mass fraction of species $a$ less than $w_a$.

## 5.3 Simulation results

In this section we will focus on the evolution of the mixing state of black-carbon-containing particles within the boundary layer. However, before we discuss the results of the aerosol mixing state in detail, we will provide a brief description of the bulk quantities of the scenario. For this 48-hour scenario, the temperature and relative humidity varied over time, as simulated by the WRF model and shown in Fig. 9. Figure 10 shows the evolution of O$_3$ and NO$_2$ profiles for the 2-day simulation period. During the daytime we observed production of O$_3$, with the highest mixing ratio of 137 ppb occurring in the afternoon of the second simulation day and peak surface O$_3$ mixing ratio of 132 ppb at 15:00 LST. NO$_2$ reached a maximum mixing ratio of 38.8 ppb found at the surface at 07:20 LST on the second day.

Figure 11 gives an overview of the evolution of aerosol bulk properties. Figure 11(a) shows the time evolution of vertical profiles of black carbon mass concentration. Black-carbon-containing particles were emitted at the surface and vertically mixed in the boundary layer by turbulent diffusion. As the stable boundary layer developed around 18:00 LST on each day, black carbon emissions accumulated within that layer with a depth of $\sim 250$ m, resulting in higher surface concentrations. A maximum black carbon mass concentration of 4.63 µg m$^{-3}$ was found at the surface at 08:20 LST as vehicle emissions began to increase and before the mixing layer began to deepen. Later in the morning the boundary layer height increased, allowing black carbon concentrations to be dispersed vertically and become well mixed.

Figures 11(b)–(d) show the bulk aerosol mass concentrations of nitrate, sulfate, and SOA. Ammonium nitrate formation was responsible for the majority of the aerosol mass, with a peak mass concentration of 27.7 µg m$^{-3}$. Maximum sulfate and SOA concentrations were 3.2 µg m$^{-3}$ and 7.7 µg m$^{-3}$, respectively, at the surface.

Figure 11(e) shows the evolution of the total number concentration. Aerosol number concentrations were impacted by emissions, coagulation, deposition and turbulent transport. During the afternoon, the boundary layer remained well-mixed with respect to number concentration. During the nighttime, the aerosol number concentration decreased with time most noticeably





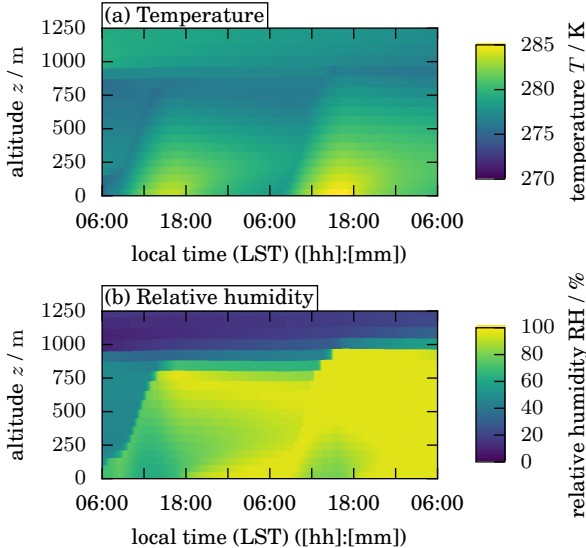

**Figure 9.** Time-height sections of (a) temperature and (b) relative humidity over the course of the 48-hour simulation.

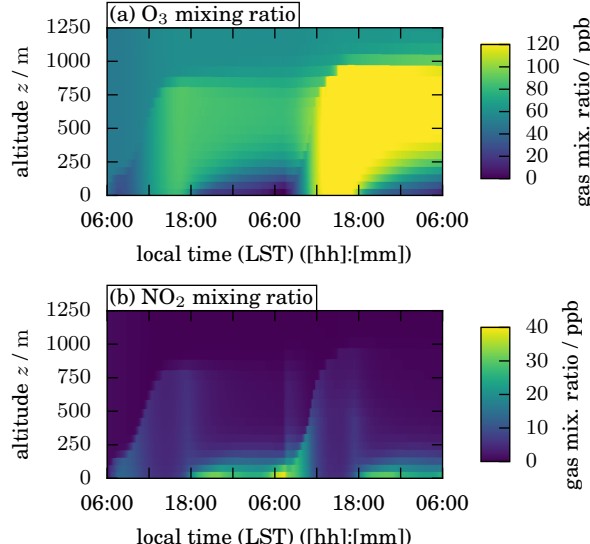

**Figure 10.** Time-height sections showing mixing ratios of (a) ozone and (b) $NO_2$.

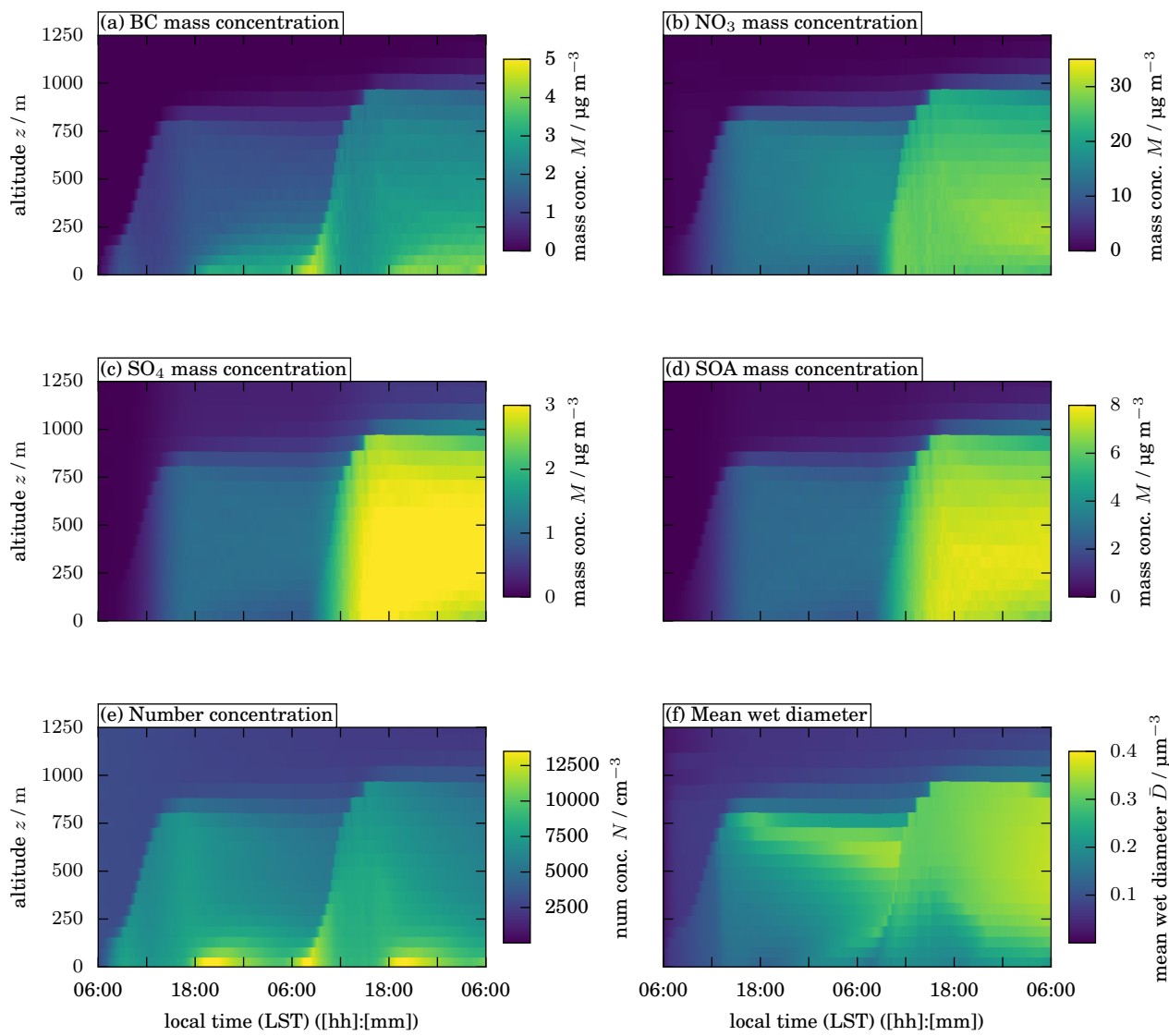

**Figure 11.** Time-height sections of aerosol mass concentrations of (a) black carbon, (b) nitrate, (c) sulfate, and (d) secondary organic aerosol. Also shown are time-height distributions of (e) total number concentration and (f) number mean wet diameter.





from the top of the stable boundary layer ($\sim 250$ m) to the top of the residual layer due to coagulation and lack of transport of surface emissions. The maximum number concentration at the surface was $\sim 14\,000$ cm$^{-3}$ at 19:30 LST, when vehicle emissions became trapped near the surface as a result of the development of the nocturnal boundary layer. Number concentrations in the stable boundary layer decreased with time overnight due to coagulation and dry deposition in combination with relatively

lower emission rates over that period.

Figure 11(f) shows how the number mean wet diameter varied with height and time. The mean wet diameter was largest in the residual layers of each night due to the particle populations containing high amounts of ammonium nitrate, which resulted in water uptake of particles in the high relative humidity environment as indicated in Fig. 9(b).

To understand how the mixing state of black-carbon-containing particles evolves in time and with respect to height, Fig. 12

shows the two-dimensional number distribution $n(D_{\mathrm{dry}}, w_{\mathrm{BC,\,dry}})$ at 06:00 and 12:00 LST on day 2 at heights of 25 m, 241 m and 552 m. Fresh emissions occur at the surface and appear as horizontal lines, with diesel emissions prescribed as $w_{\mathrm{BC,dry}} = 70\%$ and gasoline emissions as $w_{\mathrm{BC,\,dry}} = 20\%$.

At 06:00 LST on the second day, the horizontal lines representing fresh emissions were most pronounced at the surface. As a result of the stable boundary layer limiting the vertical extent of turbulent mixing, the fresh emissions were contained

to levels near the surface. By 12:00 LST, the height of the boundary layer had grown to a height of $\sim 750$ m. As a result, particles with high BC mass fraction were vertically transported to higher levels. However, the greatest number concentrations of the fresh particles were still found near the surface. Diagonal band structures of high number concentrations were a result of condensation of nitrate, which gradually shifted particles to larger diameters and lower BC mass fraction.

## 6  Code availability

The box model version of PartMC is available from http://lagrange.mechse.illinois.edu/mwest/partmc/ under the GNU General Public License (GPL) license. The version of WRF-PartMC-MOSAIC-SCM presented here, excluding coupling with MOSAIC, is available upon request from Nicole Riemer (nriemer@illinois.edu) and is available under the GNU GPL. To couple chemistry, MOSAIC may be obtained upon request from Rahul Zaveri (rahul.zaveri@pnl.gov).

## 7  Conclusions

In this paper we presented the development and application of the WRF-PartMC-MOSAIC-SCM model. This model, for the first time, resolves the aerosol composition on a per-particle level in an Eulerian single-column domain and couples the aerosol and gas phase chemistry with the meteorology.

We developed and implemented two new algorithms, a stochastic aerosol transport algorithm to treat vertical turbulent diffusion, and a stochastic aerosol dry deposition algorithm. Both model processes were verified with test cases and performed

as expected when compared to analytical solutions.

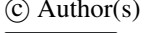



**Figure 12.** Two-dimensional number distributions $n(D_{\mathrm{dry}}, w_{\mathrm{BC,dry}})$, as defined by Eq. (74), after 24 and 30 h of simulation (left and right) at three vertical model levels (top to bottom).



To illustrate the newly coupled model capabilities, an idealized urban scenario was developed. This 48-hour simulation showed the evolution of the black carbon mixing state due to coagulation, secondary aerosol formation, particle emission, dry deposition and turbulent transport. In the presented scenario, freshly emitted diesel and gasoline particles existed in the highest concentrations near the surface where they were emitted. As particles were vertically mixed due to turbulent transport,

5  emitted particles experienced changes in composition due to coagulation with aged particles as well as due to condensation of secondary aerosol species. While we focused on the composition of black-carbon-containing particles to demonstrate the model capabilities, we do store the full composition for each computational particle, so a similar analysis can be made for other aerosol species.

Future application of the model will investigate the impact of aerosol mixing state on optical and CCN properties. This will

10  enable us to quantify the errors caused by the type of simplifying assumptions on mixing state that are common in regional and global aerosol models.

*Acknowledgements.* The authors acknowledge funding from the Office of Science (BER), U.S. Department of Energy under Grants DE-SC0003921 and DE-SC0011771. J. H. C. acknowledges support from a Computational Science and Engineering Fellowship from the University of Illinois at Urbana-Champaign. Matthew West acknowledges NSF grant CMMI-1150490.





# Appendix A: List of symbols

**Table 2.** Symbols used in this paper

| Symbol | Definition |
| --- | --- |
| $a$ | aerosol species index |
| $A$ | number of aerosol species being tracked |
| $c_a$ | conversion factor from moles of gas species $a$ to aerosol species $a$ |
| $c_w$ | conversion factor for water |
| $C_k^{\prime s}$ | deterministic real-valued particle number in grid cell $k$ at time step $s$ |
| $C_{l,m}^{G,s}$ | deterministic real-valued particle number from cell $l$ to $m$, gained by $m$ |
| $C_{l,m}^{L,s}$ | deterministic real-valued particle number from cell $l$ to $m$, lost by $l$ |
| $\gamma_{l,m}^s$ | ratio of $C_{l,m}^{L,s}$ to $C_{l,m}^{G,s}$ |
| $\dot{g}_{\mathrm{emit},i}(z,t)$ | emission rate of gas species $i$ at height $z$ and time $t$ |
| $G$ | number of gas phase species being tracked |
| $g_i(z,t)$ | concentration of gas phase species $i$ at time $t$ and height $z$ |
| $i$ | gas species or particle index |
| $I_i(\boldsymbol{\mu},\boldsymbol{g},t)$ | condensation flux of gas species $i$ onto particles with composition $\boldsymbol{\mu}$ at time $t$ |
| $I_w(\boldsymbol{\mu},\boldsymbol{g},t)$ | condensation flux for water onto particles with composition $\boldsymbol{\mu}$ at time $t$ |
| $k$ | vertical grid cell index |
| $K_{\mathrm{h}}(z,t)$ | diffusion coefficient of heat at height $z$ and time $t$ |
| $K_{\mathrm{h},k\pm\frac{1}{2}}$ | diffusion coefficient of heat at top and bottom edge of grid cell $k$ |
| $K(\boldsymbol{\mu},\boldsymbol{\mu}')$ | coagulation rate between particles $\boldsymbol{\mu}$ and $\boldsymbol{\mu}'$ |
| $\ell_d$ | removal probability of an aerosol particle due to dry deposition |
| $\boldsymbol{\mu}_i$ | particle $i$ |
| $\boldsymbol{\mu}$ | $A$-dimensional vector describing the per-species masses of an aerosol particle |
| $n(z,\boldsymbol{\mu},t)$ | aerosol number distribution at time $t$, height $z$ with constituent masses $\mu$ |
| $\dot{n}_{\mathrm{emit}}(z,\boldsymbol{\mu},t)$ | number distribution rate of aerosol emissions for particles with composition $\mu$ at time $t$ and height $z$ |
| $N_{l,m}^{G,s}$ | number concentration transported from cell $l$ to $m$, gained by $m$ |
| $N_{l,m}^{L,s}$ | number concentration transported from cell $l$ to $m$, lost by $l$ |
| $N_k^s$ | average total aerosol number concentration of grid cell $k$ |
| $\bar{N}_k^t$ | average total aerosol number concentration of grid cell $k$ from analytical solution |
| $N_{\mathrm{p}}^k$ | number of computational particles in grid cell $k$ |
| $N(z,t)$ | total aerosol number concentration at height $z$ and time $t$ |





| Symbol | Definition |
| --- | --- |
| $\Pi_k^s$ | finite set of particles of grid cell $k$ at time step $s$ |
| $\Pi_{l,m}^{\mathrm{G},s}$ | finite set of particles from cell $l$ to $m$, gained by $m$ |
| $\Pi_{l,m}^{\mathrm{L},s}$ | finite set of particles from cell $l$ to $m$, lost by $l$ |
| $\Pi_{l,m}^{\mathrm{R},s}$ | finite set of particles sampled from cell $l$ to $m$ but not removed from $l$ |
| $\Pi_{l,m}^{\mathrm{T},s}$ | finite set of particles for transfer |
| $p_{l,m}^{\mathrm{G},s}$ | loss probability from cell $l$ to $m$, gained by $m$ |
| $p_{l,m}^{\mathrm{L},s}$ | loss probability from cell $l$ to $m$, loss by $l$ |
| $p_{l,m}^{\mathrm{T},s}$ | maximum probability of loss and gain from cell $l$ to $m$ |
| $R_i(\boldsymbol{g}(z,t))$ | concentration growth rate of gas species $i$ due to gas phase chemical reactions. |
| $\rho(z,t)$ | density of air at height $z$ and time $t$ |
| $\rho_k^s$ | density of air for grid cell $k$ at time step $s$ |
| $\rho_{k\pm\frac{1}{2}}^s$ | density of air at grid cell $k$ edge at time step $s$ |
| $s$ | time step index |
| $V_{\mathrm{d},i}$ | dry deposition velocity of particle $i$ |
| $V_k^s$ | computational volume for aerosol population in grid cell $k$ at time step $s$ |
| $\Delta z_k^s$ | height difference of top and bottom edge of grid cell $k$ at time step $s$ |
| $\Delta z_{k\pm\frac{1}{2}}^s$ | height difference between center points of grid cells $k\pm1$ and $k$ at time step $s$ |

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
