# Peer review of "A single-column particle-resolved model for simulating the vertical distribution of aerosol mixing state: WRF-PartMC-MOSAIC-SCM v1.0"

_Geoscientific Model Development, 2017_

## Referee Comment (RC1) · Anonymous Referee #1 · 10 Jul 2017

General comments:

This paper presents a stochastic method to simulate in 1D the vertical turbulent diffusion and dry deposition of particles in the atmosphere. The dynamic of aerosols is resolved using the previously developed PartMC-MOSAIC model, which computes stochastically the dynamic evolution of aerosols, taking into account their mixing states. The paper is well written. The approach is interesting to model aerosol concentrations in the atmosphere without assumption on their mixing state. However, the paper needs some clarification before publication. I recommend publication after the specific com-

ments below are addressed.

Specific comments:

- Why do we need to simulate stochastically the vertical transport of particles? Gas/phase condensation/evaporation if done in a deterministic way, so why not vertical transport?

- The explicit vertical transport numerical scheme presented here may be quite expensive in terms of CPU cost for further 3D applications. Could the stochastic method be applied to implicit or semi-implicit scheme for solving vertical diffusion? Could it be less CPU expensive to use an implicit or semi-implicit scheme without a stochastic method?

- p2, l19-21: No, 3D chemical transport models that resolve additional mixing state information are not all focused on black carbon, see Zhu et al. 2016A, 2016b. To lower the computational cost of resolving the mixing state, they considered the mixing state of groups of chemical compounds, with 3D applications over Greater Paris.

Zhu S., Sartelet K., Zhang Y., Nenes A., Three-dimensional modelling of the mixing state of particles over Greater Paris J. Geophys. Res. Atmos., 121, doi:10.1002/2015JD024241, 2016a.

Zhu S., Sartelet K., Healy R., Wenger J., Simulation of particle diversity and mixing state over Greater Paris: A model-measurement inter-comparison. Faraday Discussions, 189, 547 - 566, DOI: 10.1039/C5FD00175G, 2016b.

- p5, l5-10/ Wouldn't it be more realistic to set the upper boundary condition to some value (0 or values given by a larger-scale model)? Emissions could also be taken into account at the surface as lower boundary conditions.

- p6 l5: have you tried to simulate transport before PartC and MOSAIC rather than after?

- p6 l28: How is determined the computational volume Vk from the grid cell? Please

add a short description here.

- p12. The details of the calculation are hard to follow. You should consider putting it in Appendix and adding more details, and only keep equations (37) to (39) in the paper. Equations (40), (41) could be put in place of equation (26), and the equations used for sampling can be presented with a with the whole explanation in Appendix. In the main part of the paper, you could just keep the explanation of what equations 40 and 41 have to be rewritten (because they are not independent as I understand). I assume that equation 31 is true because max() min() = 1. So it would add clarity to the paper to have more details in the Appendix about the equations through the calculation (min()pG = min(1/gamma,1) etc).

- p13. L13. What are the unsampled particles? Are they linked to Vk? Do they appear because the maximum transport term is considered rather than gain and loss? Why did they not appear in equation 25 of p10?

- p14. Using the algorithm presented here, how are we sure that in a time step the particles are not transported to levels k-2 or k+2?

- p15, p16, p17. You should consider putting the detailed algorithm in Appendix.

- How are emissions treated? At what stage of the algorithm?

- p20, Figure 6. Does this mean that the number of computational particles needs to be 10^6 for atmospheric applications?

- p22. L 17. If there is a source of emissions in the domain, is this criterion modified at the point of emissions?

- p29. L7-10. How are nitrate concentrations impacted by the aerosol mixing state?

Technical comments

- p7, figure 1: z k+1 2 should probably be replaced by z k+1/2 and z k-1 2 should probably be replaced by z k-1/2

- p26, Figure 11. Use the same scale, at least for BC, SO4 and SOA.

---

## Referee Comment (RC2) · Anonymous Referee #2 · 15 Jul 2017

Curtis et al. presents the development of a single-column particle resolved model to simulating vertical distribution of aerosol. The method used in this study is solid and the paper was properly written. I recommend its publication after my following comments are addressed. Comments:

1. Vertical transport is more generally used to represent vertical movement and distribution of aerosols. But not only turbulent diffusion and dry deposition affect vertical distribution, why only they are considered?

[Figure]

2. WRF already has vertical transport schemes. Why this study uses new and different equations when WRF is coupled?

3. WRF has Asymmetric Convective Model, version 2, (ACM2) to include both an eddy diffusion scheme and the nonlocal scheme to better represent the rise and fall of the convective boundary layer. Has this been considered in this model?

4. The abstract is rather simple. Only what have been done were presented but no results were shown.

5. If aerosol mixing state is used to refer distribution of chemical species. Then, all current models are able to and predicting aerosol mixing state. What makes this study different? Mixing state is better used for how particle components are distributed in each particle, homogeneous, core-shell or else. But it is not discussed in this study.

6. Too much detailed information in sections 2 and 3. They should be greatly reduced by put information to appendix. Very less readers would be interested in the algorithms.

7. Point source emissions are important in vertical distribution calculation of particle? Why this study did not consider that? How would that change the results?

8. The tested case only shows the concentrations of PM components. It is not clear how mixing state is changed or simulated as the title emphasizes it.

---

## Author Comment (AC1) · 3 Sep 2017

**1 Responses to Reviewer #1**

We thank the reviewer for taking the time to review our paper and for the constructive comments. The page and line numbers that we quote for indicating where we changed the manuscript refer to the revised marked-up version.

**(1.1)** Why do we need to simulate stochastically the vertical transport of particles? Gas/phase condensation/evaporation if done in a deterministic way, so why not vertical transport?

> The reason for using stochastic methods for processes such as transport and coagulation is their computational efficiency. The non-stochastic alternative to simulate particle transport would be to track and update the exact location of each simulated particle. This can be done, but would be computationally more expensive than the stochastic sampling. By using a stochastic sampling approach, only a fraction of particles will have their grid cell positions updated each time step. In many cases, this transported fraction is rather small. In contrast, the gas and aerosol chemistry is simulated deterministically, so that we could re-use existing chemistry libraries (MOSAIC). To clarify the importance of a stochastic method for efficiency, we made the following change to the manuscript:
>
> - In Section 3.1, page 6, line 32, we added: "We use a stochastic sampling approach for moving particles between grid boxes, rather than tracking the exact location of each simulated particle. This is done for computational efficiency. By using a stochastic sampling approach, only a fraction of particles will have their grid cell positions updated in each time step. In many cases, this transported fraction is rather small."

**(1.2)** The explicit vertical transport numerical scheme presented here may be quite expensive in terms of CPU cost for further 3D applications. Could the stochastic method be applied to implicit or semi-implicit scheme for solving vertical diffusion? Could it be less CPU expensive to use an implicit or semi-implicit scheme without a stochastic method?

> We chose an explicit method because this simplified the parallel (MPI) implementation of the model where the vertical domain is divided onto many cores. The explicit method has the benefit of requiring communication with only nearest cores. This is in contrast to an implicit method where particles potentially are transported many grid cells away with those grid cells being located on distant processors.
>
> There is no theoretical limitation preventing the use of a semi-implicit or implicit scheme. However, several aspects would require consideration. A major disadvantage of semi-implicit or implicit methods is that the solution process results in an inverse of the triadiagonal matrix, which produces a dense matrix of sampling probabilities. The implication of a dense matrix is that particles in grid cell $k$ must be sampled to all other grid cells in the domain. To relax this, a threshold could be applied to ignore very small probabilities. While implicit methods have the advantage of stability and allow for larger time steps, a time step would need to be chosen to maintain a diagonally dominant matrix to avoid any negative values within the probability matrix.
>
> It is also important to evaluate the potential CPU cost savings for the implicit or semi-implicit methods. For our case, the use of a such methods would not save a significant amount of CPU time. This is because the CPU expense of particle transport scales to the number of computational particles transported. The evaluation of the diffusion equation to determine probabilities are computed only once per grid cell per model time step $\Delta t$. The sampling of particles is more costly as particles are sampled and moved to other grid cells. This moving also involves a MPI communication step if the source grid cell and destination grid cell are on different cores. Any method that allows for a larger time step results in more particles being sampled and increases computational cost. Computational cost for transport is primarily dependent on the amount of mixing and grid resolution.

- In Section 3.1.1, we added (page 8, line 18): "An explicit, second-order accurate discretization scheme was selected for this work. While an explicit method simplifies the parallel implementation because it only requires communication between neighboring grid cells, there exists no theoretical reason why other numerical schemes may not be used, including higher-order, semi-implicit, or implicit methods."

We also considered the cost of transport in comparison to other processes simulated by the model. While it is possible that we may introduce some future efficiency improvements to particle transport, it turns out that transport is relatively inexpensive. The chemistry is dominated by condensation/evaporation computed per particle per grid cell per time step and represents the majority of the CPU cost of the model. We highlighted the computational cost of the model with the following changes to the manuscript:

- We inserted a brief new subsection Section 5.4 (page 25). This subsection details the computational costs of the model for each of the four components as listed in Section 3.

Stochastic methods allow this model to be computationally feasible. The deterministic approach, which is particle tracking, requires the updating of each particle per time step. This was addressed in the response to comment (1.1).

**(1.3)** p2, l19-21. No, 3D chemical transport models that resolve additional mixing state information are not all focused on black carbon, see Zhu et al. 2016A, 2016b. To lower the computational cost of resolving the mixing state, they considered the mixing state of groups of chemical compounds, with 3D applications over Greater Paris.

- Zhu S., Sartelet K., Zhang Y., Nenes A., Three-dimensional modelling of the mixing state of particles over Greater Paris J. Geophys. Res. Atmos., 121, doi:10.1002/2015JD024241, 2016a.

- Zhu S., Sartelet K., Healy R., Wenger J., Simulation of particle diversity and mixing state over Greater Paris: A model-measurement inter-comparison. Faraday Discussions, 189, 547 - 566, DOI: 10.1039/C5FD00175G, 2016b.

We agree that not all mixing state models only focus on representing black carbon mixing state. We changed the manuscript accordingly as follows:

- In the Introduction, we clarified by adding (page 2, line 20): "While some chemical transport models focus on representing black carbon mixing state [Matsui et al., 2013, Matsui, 2016], other models have allowed for more general mixing state representations [Zhang et al., 2014, Zhu et al., 2016]."

**(1.4)** p5, l5-10/ Wouldn't it be more realistic to set the upper boundary condition to some value (0 or values given by a larger-scale model)? Emissions could also be taken into account at the surface as lower boundary conditions.

Our study and anticipated future studies are limited to the boundary layer such that the upper boundary condition is not particularly critical. The version of WRF that was used (v3.3.1) in the coupling pre-dates the version of WRF-Chem that allows for the setting of upper boundary conditions with the namelist variable `have_bcs_upper`. Current versions of WRF-Chem do not have a specified upper boundary conditions turned on by default. This option is something that we will consider in the future to remain consistent with WRF-Chem development.

We could consider emissions as lower boundary conditions if they were strictly limited to the surface. However, since we wanted to allow for the possibility of elevated point sources, we did not include them as lower boundary conditions. Instead, emissions may be released in any model

layer. We made the following change to the manuscript to emphasize that emissions are not limited to the surface:

- In Section 2, page 4, after Equation (2) we added: "the number distribution rate of aerosol emissions which can be specified at any height".

**(1.5)** p6 l5: have you tried to simulate transport before PartMC and MOSAIC rather than after?

We have not investigated the effect of changing the order of the operator splitting sequence. As the time step becomes smaller, any order of operations will converge, and we expect that the operator splitting to have minor effects for the small time steps we are using. However, this is an aspect we would like to give future consideration to, in addition to ordering within the chemistry module itself. Studies such as Santillana et al. [2016] investigated operator splitting in GEOS-Chem and found the operator splitting error to be smaller than the error in transport, albeit the focus was on numerical diffusion resulting from advection.

**(1.6)** p6 l28: How is determined the computational volume $V_k$ from the grid cell? Please add a short description here.

The computational volume $V_k$ is the ratio of the number of computational particles contained in a grid cell to the number concentration of a grid cell. The computational volume is adjusted as necessary to maintain roughly the desired number of computational particles. The process of rebalancing the particle population and adjusting the computational volume is described in Section 3.1.5. We made the following change to the manuscript:

- To make the relationship clear, we added on page 7, line 9: "The value of the computational volume is the ratio of the number of computational particles contained in the grid cell over the number concentration of the grid cell."

**(1.7)** The details of the calculation are hard to follow. You should consider putting it in Appendix and adding more details, and only keep equations (37) to (39) in the paper. Equations (40), (41) could be put in place of equation (26), and the equations used for sampling can be presented with a with the whole explanation in Appendix. In the main part of the paper, you could just keep the explanation of what equations 40 and 41 have to be rewritten (because they are not independent as I understand). I assume that equation 31 is true because max() min() = 1. So it would add clarity to the paper to have more details in the Appendix about the equations through the calculation (min()pG = min(1/gamma,1) etc).

We agree that the series of equation describing the sampling was difficult to follow. To make it more accessible to the reader, we made the following changes to the manuscript:

- We removed Equation 25.
- We combined Section 3.1.2 and 3.1.3 by reordering equations to make this section more sequential instead of presenting one mathematical idea and a nearly equivalent approach. This involved removing Equations 37–39 and replacing them by inserting Equations 42–52.
- In response to a comment by Reviewer 2, we also moved a number of derivation equations from 3.1.1 to Appendix B.

**(1.8)** p13. L13. What are the unsampled particles? Are they linked to $V_k$? Do they appear because the maximum transport term is considered rather than gain and loss? Why did they not appear in equation 25 of p10?

Unsampled particles are particles that were not sampled at any point in the time step. These particle sets appear as we multinomial sample particles for the transfer sets in each direction. Equation 25 and 52 are equivalent but dealt with different perspectives regarding the creation of particle sets. To clear up this confusion, we made the following change:

- As part of the simplification of the equations found in Section 3.1.2 and 3.1.3 in response to Reviewer comment (1.7), we removed Equation 25.

**(1.9)** p14. Using the algorithm presented here, how are we sure that in a time step the particles are not transported to levels $k - 2$ or $k + 2$?

To make sure that this does not happen, we introduced the sub-cycling time step within a general model time step (as detailed in Section 3.1.3). During a general model time step, particles may be transported more than one grid cell away. If turbulent mixing is small, particles will not travel far in a general model time step and the sub-cycling step can be large. In the cases where turbulent mixing is large, the sub-cycling time step (as detailed in Section 3.1.3) is made sufficiently small so that particles may only go to $k - 1$ and $k + 1$. In this case, particles can be transported to $k - 2$ and $k + 2$, or further, in a general time step. This comment is an interesting point and has been emphasized with the following change:

- In Section 3.1.3, page 13, line 26, we added the following: "Within each sub-cycle time step $\Delta t_{\mathrm{T}}$, particles will only transition between immediate grid cell neighbors. However, within a full model time step $\Delta t$, particles may be transported more than one grid cell away."

**(1.10)** p15, p16, p17. You should consider putting the detailed algorithm in Appendix.

We agree with this suggestion regarding shortening this section of the paper. We made the following changes:

- We moved Algorithm 1 and 2 to Appendix C1.
- We moved Algorithm 3 to Appendix C2.

**(1.11)** How are emissions treated? At what stage of the algorithm?

Emissions are treated stochastically as described in Riemer et al. [2009]. Within a given time step, a finite number of particles is emitted that are sampled from aerosol emission distributions. This process is simulated in the PartMC model and highlighted in Section 3 where the model operating splitting is discussed. To make it more clear how the PartMC module works, we made the following change:

- In Section 3, we added details regarding stochastic particle emission and coagulation (page 6, line 15): "Emissions are simulated by stochastically sampling a finite number of particles at each time step, approximating the continuum emission distribution. Coagulation is efficiently simulated using a fixed time step method and a binned acceptance procedure. These approaches are further described in Riemer et al. [2009], DeVille et al. [2011], and Michelotti et al. [2013]."

**(1.12)** p20, Figure 6. Does this mean that the number of computational particles needs to be $10^6$ for atmospheric applications?

No, the purpose of this figure is to show convergence properties of the method to the expected value. In the case of particle transport, there is a balance between physical grid cell size $\Delta z$ and the number of computational particles $N_\mathrm{p}$.

This result is not sufficient to determine an acceptable number of computational particles for simulations in general. This should be determined on a case by case basis. The number of computational particles that is needed for a given scenario is determined by other aspects of the case, such as the resolution of the size distribution and representation of different types of particles from different emission sources. To clarify this point we made the following change:

- In Section 4, page 15, line 10, we added text to emphasize that the verification section is only for verification: "These verification scenarios are motivated by atmospheric applications, but have somewhat artificial grid structures (see Figure 4), chosen to enable smooth refinement studies. Therefore, the numerical values in this section will differ from those in actual atmospheric case studies.

**(1.13)** p22. L 17. If there is a source of emissions in the domain, is this criterion modified at the point of emissions?

No, the criterion is always applied, regardless if there are emissions or not. When particles are emitted into a given grid cell, the number of computational particles will increase. When the total number of computational particles reaches a factor of two greater than the number of ideal computational particles, we halve the particle population as mentioned in Section 3.1.4. We added the following text:

- In Section 5.1, page 20, line 20, we added an explanation of how the computational particle number changes during the simulation: "The number of computational particles within each level fluctuated during the simulation due to emission, coagulation, transport and dry deposition, and was restricted to a range between half and double the initial number of particles (12 500 to 50 000) to maintain accuracy while avoiding higher computational costs as described in Sec. 3.1.4."

**(1.14)** p29. L7-10. How are nitrate concentrations impacted by the aerosol mixing state?

This is a very interesting question, however we cannot quantify this impact yet, as we have not yet performed a simulation where mixing state is *not* resolved. While this was beyond the scope of this model development paper, it will certainly be of future interest to quantify the effects of simulating mixing state on physical processes and properties. We made the following addition to the Conclusion section to comment on possible future applications:

- On page 27, line 25, we added: "Future applications of the model include quantifying the impact of aerosol mixing state on secondary aerosol formation and on climate-relevant aerosol properties, such as aerosol absorption and CCN concentration, and to compare these findings to existing studies [Matsui et al., 2013, Zhang et al., 2014, Zhu et al., 2016]."

**(1.15)** p7, figure 1: $z_{k+1/2}$ should probably be replaced by $z_{k+\frac{1}{2}}$ and $z_{k-1/2}$ should probably be replaced by $z_{k-\frac{1}{2}}$

Thanks for pointing out this inconsistency. We made the corresponding change:

- We adjusted the labels for the grid cell edges of $k$ in Figure 1.

**(1.16)** p26, Figure 11. Use the same scale, at least for BC, SO4 and SOA.

We tried applying the same scales for BC, SO4 and SOA, but concluded that even though they are of similar magnitude, certain features, such as the concentrations found in the nocturnal boundary layer, are lost when the color scales are all equal. However to avoid any possible confusion, we made the following change:

- We added to Figure 11 caption: "Note that color scales for each chemical species differ."

**References**

R. E. L. DeVille, N. Riemer, and M. West. Weighted flow algorithms (WFA) for stochastic particle coagulation. *J. Geophys. Res.*, 230(23):8427–8451, 2011. doi: 10.1016/j.jcp.2011.07.027.

H Matsui. Black carbon simulations using a size-and mixing-state-resolved three-dimensional model: 1. radiative effects and their uncertainties. *Journal of Geophysical Research: Atmospheres*, 121(4), 2016. doi: 10.1002/2015JD023998.

H Matsui, M Koike, Y Kondo, N Moteki, J D Fast, and R A Zaveri. Development and validation of a black carbon mixing state resolved three-dimensional model: Aging processes and radiative impact. *Journal of Geophysical Research: Atmospheres*, 2013. doi: 10.1029/2012JD018446.

M. D. Michelotti, M. T. Heath, and M. West. Binning for efficient stochastic multiscale particle simulations. *Multiscale Modeling & Simulation*, 11(4):1071–1096, 2013. doi: 10.1137/130908038.

N. Riemer, M. West, R. A. Zaveri, and R. C. Easter. Simulating the evolution of soot mixing state with a particle-resolved aerosol model. *J. Geophy. Res.*, 114:D09202, 2009. doi: 10.1029/2008JD011073.

Mauricio Santillana, Lin Zhang, and Robert Yantosca. Estimating numerical errors due to operator splitting in global atmospheric chemistry models: Transport and chemistry. *Journal of Computational Physics*, 305: 372–386, 2016.

H Zhang, SP DeNero, DK Joe, H-H Lee, S-H Chen, J Michalakes, and MJ Kleeman. Development of a source oriented version of the WRF/Chem model and its application to the California regional PM 10/PM 2.5 air quality study. *Atmos. Chem. Phys.*, 14(1), 2014. doi: 10.5194/acp-14-485-2014.

Shupeng Zhu, Karine Sartelet, Yang Zhang, and Athanasios Nenes. Three-dimensional modeling of the mixing state of particles over Greater Paris. *Journal of Geophysical Research: Atmospheres*, 121(10):5930–5947, 2016. ISSN 2169-8996. doi: 10.1002/2015JD024241. URL http://dx.doi.org/10.1002/2015JD024241. 2015JD024241.

---

## Author Comment (AC2) · 3 Sep 2017

**1 Responses to Reviewer #2**

We greatly appreciate the reviewer's comments. The page and line numbers that we quote for indicating where we changed the manuscript refer to the revised marked-up version. Our responses are as follows:

**(2.1)** Vertical transport is more generally used to represent vertical movement and distribution of aerosols. But not only turbulent diffusion and dry deposition affect vertical distribution, why only they are considered?

In the single column model, we only consider vertical advection due to density changes, which is handled in the WRF timestep component. The grid in the WRF model is constant in its native hydrostatic-pressure coordinate $\eta$ but not constant in height $z$. Any pressure-induced changes therefore cause the $z$ grid to shift and this is entirely handled by the WRF model grid transformation. It is important to note that particles will not shift grid cells during this process.

In the case of considering large scale vertical advection, similar equations could be derived to create the corresponding transport probabilities. This aspect will be important in the future as the model is extended to be fully three-dimensional where large-scale vertical advection will exist. To explain the treatment of the vertical advection term, we made the following change:

- In Section 3, page 6, line 8, we added: "The WRF model is discretized using a terrain-following hydrostatic-pressure coordinate system $\eta$ which is constant in time. The aerosols and gas species are evolved in the transport step ($\Phi_{\Delta t}^{\mathrm{Trans}}$) on a geometric height coordinate system $z$ that is computed from the geopotential field and changes over time due to column pressure changes. The vertical advection terms found in Eqs. (2) and (3) are entirely due to pressure-induced grid changes and do not cause particles to be transported across grid cell edges. At every time step the $z$ grid is moved by WRF, and this accounts for vertical advection."

**(2.2)** WRF already has vertical transport schemes. Why this study uses new and different equations when WRF is coupled?

The WRF vertical transport scheme operates on scalar quantities such as mass mixing ratios. The method we presented is similar to WRF but includes an additional step where we discretize the equations to finite particle number. This is necessary because WRF-PartMC-MOSAIC uses a particle-resolved approach where individual particles, and not scalars, are transported. Equations are presented in terms of sampling and probabilities as the stochastic approach avoids the computational cost of tracking the exact vertical position of each particle. We added the following to the manuscript:

- In Section 3.1, page 7, line 18, we added the following to clearly explain the difference between WRF/WRF-Chem and our model: "This is in contrast to conventional models which transport scalar variables such as mass mixing ratios. Therefore, the discretization process requires an additional step to transport particles."

**(2.3)** WRF has Asymmetric Convective Model, version 2, (ACM2) to include both an eddy diffusion scheme and the nonlocal scheme to better represent the rise and fall of the convective boundary layer. Has this been considered in this model?

When we started the work that is presented in our manuscript, the ACM2 scheme was not yet supported in WRF-Chem [Pleim, 2011], and therefore we did not consider it.

For simplicity, our current model formulation requires local K theory so a local PBL scheme is required as particles may only move to nearest neighbors within a sub cycle time step. This

constraint could be removed in the future for use with the ACM2 scheme or any other non-local scheme. This will require rewriting the transport equations to determine particle transfer probabilities from any grid cell to any other grid cell such as by using a transilient matrix. For example, the non-local term in ACM2 involves the upward transport of the mass mixing ratio from the lowest grid cell $C_1$ to all over grid cells above. This would involve sampling for the particle population of the lowest grid to other grid cells above it.

In the future, we will look to include ACM2 because of its good performance, to be consistent with other chemical transport models (e.g., CMAQ), and to allow for additional flexibility in the selection of boundary layer schemes. We made the following comments in the manuscript about PBL schemes:

- In Section 5.1, page 20, line 16, we added to clarify the valid choices of boundary layer schemes: "The presented model formulation requires the use of local boundary layer schemes such as MYJ and Mellor-Yamada-Nakanishi-Niino (MYNN) [Nakanishi and Niino, 2006, 2009]. Non-local schemes such as Asymmetric Convection Model 2 Scheme (ACM2) [Pleim, 2007] may be included in future work."

- In Section 7, page 27, line 14, we added a comment to emphasize that the methods presented in this paper could be extend to include ACM2 in the future: "Potential future model development includes the implementation of other numerical methods for turbulent diffusion, such as higher-order and/or semi-implicit schemes, and non-local boundary layer schemes such as ACM2."

**(2.4)** The abstract is rather simple. Only what have been done were presented but no results were shown.

We submitted this paper under GMD's "model development paper" category where the main goal is to "describe both the underlying scientific basis and purpose of the model and overview the numerical solutions employed." (see https://www.geosci-model-dev.net/by_ms_types.html). The contribution of this paper therefore consists of the model development in terms of the numerical methods and the verification of the model algorithms, which is the reason why the abstract focuses on these aspects.

**(2.5)** If aerosol mixing state is used to refer distribution of chemical species. Then, all current models are able to and predicting aerosol mixing state. What makes this study different? Mixing state is better used for how particle components are distributed in each particle, homogeneous, core-shell or else. But it is not discussed in this study.

The reviewer points out an important source of confusion. The uniqueness of the particle-resolved approach used in PartMC-MOSAIC is that we track the composition of individual particles, i.e., we do not need to make an assumption about how chemical species are distributed amongst different particles. We are using the term mixing state in the sense of Winkler [1973] who noted that "the same net composition of an aerosol can be caused by an infinite variety of different internal distributions of the various compounds." In this sense, mixing state is a property of the aerosol *population*, not of individual particles. Traditional aerosol models that use sections or modes are based on the assumption that within one bin or mode all particles are internally mixed, and hence do not resolve (or not fully resolve) mixing state. Importantly, this is distinct from the issue of how the chemical species are arranged *within* the particles (e.g., homogeneous, core-shell, or else). With PartMC-MOSAIC, we do not predict the arrangement of chemical species within the particles.

To clarify this we made the following changes in the paper (page 1, line 17): "For the purposes of this paper we use the term "aerosol mixing state" to refer to the distribution of chemical species across the aerosol *population* [Riemer and West, 2013, Winkler, 1973]. This is distinct

from the use of the term "mixing state" for the arrangement of components *within* a particle (e.g., homogeneous mixture or core-shell arrangements)."

**(2.6)** Too much detailed information in sections 2 and 3. They should be greatly reduced by put information to appendix. Very less readers would be interested in the algorithms.

We agree with this comment and reorganized Sections 2 and 3 to follow the suggestions of the reviewer:

- The finite volume discretization derivation (Equations 14–21 in the original manuscript) was moved to Appendix B. These equations are important steps to arrive at the final model expression (Equation 22 in the original manuscript) but may not be of primary interest to readers.
- We combined Section 3.1.2 and 3.1.3.
- We moved the stochastic algorithms for transport and dry deposition to Appendix C.

**(2.7)** Point source emissions are important in vertical distribution calculation of particle? Why this study did not consider that? How would that change the results?

Point source emissions are indeed important, and will be included in our framework once we extended it to 3D. However, for the 1D column model, our underlying assumption is horizontal homogeneity, and hence we consider emissions at the surface, representing an extended area source, but not a point source. We added the following text to clarify this:

- In Section 5.1, page 20, line 5, we added: "The model allows for the inclusion of aerosol emissions within any grid cell in the column and has flexibility in the choice of the parameters of the size distribution as well as the particle composition of the emitted particles."
- In Section 5.1, page 20, line 12, we added: "We consider this set of simplified surface emissions, as the underlying assumption of the 1D column setup is horizontal homogeneity. "

**(2.8)** The tested case only shows the concentrations of PM components. It is not clear how mixing state is changed or simulated as the title emphasizes it.

Our response to this comment is related to the response to comment (2.5). Since we define mixing state as detailed in our response to (2.5), the key figure that illustrates mixing state is Figure 12. This figure shows two-dimensional number distributions in terms of particle diameter and particle BC mass fraction for three different altitudes in the boundary layer. It conveys the information of how BC is distributed over the populations. We see that for a given size (e.g. 100 nm), the particles can have a wide range of different BC mass fractions. It is important to note that we know these distributions for all aerosol components, not only BC. To clarify this, we made the following change to the manuscript:

- At the very end of Section 5.3 we added a new final paragraph: "While Figure 12 only shows the BC-diameter distribution of the aerosol, the simulation results contain the full high-dimensional distribution over all constituent species, thus permitting the calculation of any desired mixing state measures or visualizations."

**References**

Mikio Nakanishi and Hiroshi Niino. An improved Mellor-Yamada level-3 model: Its numerical stability and application to a regional prediction of advection fog. *Boundary-Layer Meteorology*, 119(2):397–407, 2006.

Mikio Nakanishi and Hiroshi Niino. Development of an improved turbulence closure model for the atmospheric boundary layer. *Journal of the Meteorological Society of Japan. Ser. II*, 87(5):895–912, 2009.

Jonathan E Pleim. A combined local and nonlocal closure model for the atmospheric boundary layer. Part I: Model description and testing. *Journal of Applied Meteorology and Climatology*, 46(9):1383–1395, 2007.

Jonathan E Pleim. Comment on "Simulation of surface ozone pollution in the central gulf coast region using WRF/Chem model: sensitivity to PBL and land surface physics". *Advances in Meteorology*, 2011, 2011.

N. Riemer and M. West. Quantifying aerosol mixing state with entropy and diversity measures. *Atmos. Chem. Phys.*, 13(22):11423–11439, 2013. doi: 10.5194/acp-13-11423-2013. URL http://www.atmos-chem-phys.net/13/11423/2013/.

Peter Winkler. The growth of atmospheric aerosol particles as a function of the relative humidity—II. an improved concept of mixed nuclei. *Journal of Aerosol Science*, 4(5):373–387, 1973.